# Retrieval of total column and surface NO₂ from Pandora zenith-sky measurements

Xiaoyi Zhao[1], Debora Griffin[1], Vitali Fioletov[1], Chris McLinden[1], Jonathan Davies[1], Akira Ogyu[1], Sum Chi Lee[1], Alexandru Lupu[1], Michael D. Moran[1], Alexander Cede[2,3], Martin Tiefengraber[3,4], Moritz Müller[3,4]

[1]Air Quality Research Division, Environment and Climate Change Canada, Toronto, M3H 5T4, Canada.
[2]NASA Goddard Space Flight Center, Greenbelt, MD 20771, USA.
[3]LuftBlick, Kreith 39A, 6162 Mutter, Austria.
[4]Department of Atmospheric and Cryospheric Sciences, University of Innsbruck, Innsbruck, Austria.

*Correspondence to*: Xiaoyi Zhao (xiaoyi.zhao@canada.ca)

**Abstract.** Pandora spectrometers can retrieve nitrogen dioxide (NO₂) vertical column densities (VCDs) via two viewing geometries: direct-sun and zenith-sky. The direct-sun NO₂ VCD measurements have high quality (0.1 DU accuracy in clear-sky conditions) and do not rely on any radiative transfer model to calculate air mass factors (AMFs); however, they are not available when the sun is obscured by clouds. To perform NO₂ measurements in cloudy conditions, a simple but robust NO₂ retrieval algorithm is developed for Pandora zenith-sky measurements. This algorithm derives empirical zenith-sky NO₂ AMFs from coincident high-quality direct-sun NO₂ observations. Moreover, the retrieved Pandora zenith-sky NO₂ VCD data are converted to surface NO₂ concentrations with a scaling algorithm that uses chemical-transport-model predictions and satellite measurements as inputs. NO₂ VCDs and surface concentrations are retrieved from Pandora zenith-sky measurements made in Toronto, Canada, from 2015 to 2017. The retrieved Pandora zenith-sky NO₂ data (VCD and surface concentration) show good agreement with both satellite and in situ measurements. The diurnal and seasonal variations of derived Pandora zenith-sky surface NO₂ data also agree well with in situ measurements (diurnal difference within ± 2 ppbv). Overall, this work shows that the new Pandora zenith-sky NO₂ products have the potential to be used in various applications such as future satellite validation in moderate cloudy scenes and air quality monitoring.

## 1 Introduction

Nitrogen dioxide (NO₂) is an important air pollutant and plays a critical role in tropospheric photochemistry (e.g., ECCC, 2016; EPA, 2014). It is primarily emitted from combustion processes such as fossil fuel combustion (e.g., traffic, electricity generation from power plants) and biomass burning, as well as from lightning. NO₂ is a nitrate aerosol precursor, and it also contributes to acid deposition and eutrophication (ECCC, 2016). Exposure to NO₂ can lead to adverse health effects, such as irritation of the lungs, a decrease in lung function, and an increase in susceptibility to allergens for people with asthma (EEA, 2017; WHO, 2017).

As surface $NO_2$ concentrations are regulated by many environmental agencies (e.g., Environment and Climate Change Canada and U.S. Environment Protection Agency), in situ $NO_2$ measurements are commonly carried out by many national monitoring networks, such as the National Air Pollution Surveillance (NAPS, https://www.canada.ca/en/environment-climate-change/services/air-pollution/monitoring-networks-data/national-air-pollution-program.html) network in Canada, which was

established in 1969. The in situ methods used to measure surface $NO_2$ have evolved over the years; for example, luminol chemiluminescence (e.g., Kelly et al., 1990; Maeda et al., 1980; Wendel et al., 1983), long-path differential optical absorption spectroscopy (e.g., Platt, 1994), photolytic conversion/chemiluminescence (e.g., Gao et al., 1994; Ryerson et al., 2000), and laser-induced fluorescence (e.g., Thornton et al., 2000) are all found to be reliable methods with an uncertainty within 10 % at the 1 ppbv and higher concentration levels (McClenny, 2000). Currently, the in situ approach used by NAPS for surface $NO_2$

air quality monitoring is the photolytic conversion/chemiluminescence technique, which converts $NO_2$ to NO and subsequently detects the NO by chemiluminescence reaction (McClenny, 2000; NRC, 1992). This in situ monitoring measurements provides good measurements at ground level (0.4 ppbv accuracy), but $NO_2$ is not uniformly mixed through the atmosphere, and not even within the atmospheric boundary layer due to emission and removal processes taking place at the surface.

Total vertical column $NO_2$ can be measured by many ground-based UV-visible remote sensing instruments using direct-sun,

zenith-sky, or off-axis spectroscopy techniques (Cede et al., 2006; Drosoglou et al., 2017; Herman et al., 2009; Lee et al., 1994; Noxon, 1975; Piters et al., 2012; Roscoe et al., 2010; Tack et al., 2015; Vaughan et al., 1997). These measurements are of high quality and good precision, and have been widely used for atmospheric chemistry studies (e.g. Adams et al., 2012; Hendrick et al., 2014) and satellite validations (e.g., Celarier et al., 2008; Drosoglou et al., 2018; Irie et al., 2008; Wenig et al., 2008). Among all these different viewing geometries, direct-sun measurements are of high accuracy, and are not dependent on

radiative transfer models (RTMs) to calculate air mass factors (AMFs) (Herman et al., 2009) or on knowledge of other atmospheric constituents. Zenith-sky observations have been widely used for stratospheric ozone and $NO_2$ observations, particularly under cloudy conditions when direct-sun measurements are unreliable (note that zenith-sky observations use scattered sunlight and are less sensitive to clouds, e.g., Zhao et al. (2019)). Off-axis measurements have good sensitivity in the boundary layer and could provide tropospheric trace gas profiles and surface concentrations (Frieß et al., 2011; Hendrick et

al., 2014; Kramer et al., 2008; Wagner et al., 2011), but they are more sensitive to cloud cover than zenith-sky measurements. The Pandora sun spectrometer is a new instrument developed to measure vertical column densities (total columns) of trace gases in the atmosphere using sun and sky radiation in the UV-visible part of the spectrum (Herman et al., 2009). One of its primary data products is $NO_2$ total vertical column density (VCD) from the direct-sun viewing mode, where VCD represents the vertically integrated number of molecules per unit area and is reported in units of molec $cm^{-2}$ or Dobson Unit (1 DU =

$2.6870 \times 10^{16}$ molec $cm^{-2}$). The Pandora direct-sun $NO_2$ VCD products have been validated through many field campaigns (Flynn et al., 2014; Lamsal et al., 2017; Martins et al., 2016; Piters et al., 2012; Reed et al., 2015), ground-based comparisons (Herman et al., 2009; Wang et al., 2010), and satellite validations (Ialongo et al., 2016; Lamsal et al., 2014).

Since their introduction in 2006, Pandora spectrometers have been deployed at more than 50 sites globally. The Pandora no. 103 instrument used in this study has been deployed in Toronto, Canada since 2013 to perform direct-sun measurements (Zhao

et al., 2016). Since 2015, the observation schedule of Pandora no. 103 has been modified to perform alternating direct-sun and zenith-sky measurements. Knepp et al. (2017) assessed Pandora's capability to derive stratospheric $NO_2$ using zenith-sky viewing geometry (in twilight periods), but their study was limited to slant column densities (SCDs). At this time, there are no standard Pandora zenith-sky $NO_2$ VCD data products available. As one goal of this work, we have focused on developing a
new $NO_2$ retrieval algorithm for zenith-sky measurements to expand Pandora $NO_2$ measurements into cloudy scenes.

In addition to retrieval of zenith-sky total column $NO_2$, another goal of this work is to derive surface $NO_2$ concentration from total column measurements. Surface $NO_2$ has been a focus of scientific studies due to its strong correlation with air quality (AQ) and health issues (ECCC, 2016), with $NO_2$ as one of the three components (along with ozone and $PM_{2.5}$) used to compute the Air Quality Health Index (AQHI: Stieb et al., 2008) in Canada's AQ public awareness programs. Efforts to link total
column $NO_2$ with its surface concentrations have been made by many researchers (Flynn et al., 2014; Knepp et al., 2015; Kollonige et al., 2017; Lamsal et al., 2008, 2014; McLinden et al., 2014). For example, Knepp et al. (2015) proposed a method to estimate $NO_2$ surface mixing ratios from Pandora direct-sun total column $NO_2$ via application of a planetary boundary-layer (PBL) height correction factor. Kollonige et al. (2017) adapted this method and compared Pandora direct-sun surface $NO_2$ and OMI surface $NO_2$. They concluded that the two main sources of error for the conversion of the total column $NO_2$ to surface
$NO_2$ are (1) poor weather conditions (e.g., cloud cover and precipitation) and (2) PBL height estimation, both of which affect the $NO_2$ column-surface relationship and instrument sensitivities to boundary layer $NO_2$. Thus, in this work we present a simple but robust algorithm for deriving surface $NO_2$ concentration from Pandora zenith-sky measurements, which has several advantages such as the ability (1) to extend Pandora $NO_2$ measurements to cloudy conditions and (2) to provide more accurate surface $NO_2$ concentration estimates that are less sensitive to PBL height. This work also provides reliable total column $NO_2$
measurements in cloudy conditions and could be used in satellite validations in partially cloudy scenes.

This paper is organized as follows. Section 2 describes the measured and modelled $NO_2$ data used in this study. In Section 3, the empirical AMFs for Pandora zenith-sky $NO_2$ measurements are derived using high-quality Pandora direct-sun total column $NO_2$ data. These empirical AMFs and the Network for the Detection of Atmospheric Composition Change (NDACC) AMFs (Hendrick et al., 2011; Sarkissian et al., 1995; Van Roozendael et al., 1998; Van Roozendael and Hendrick, 2009; Vaughan et
al., 1997) are both applied to Pandora zenith-sky total column $NO_2$ retrievals to help evaluate the performance of the empirical AMFs. Also, the retrieved Pandora zenith-sky total column $NO_2$ data are evaluated by comparison with satellite measurements. In Section 4, the zenith-sky total column $NO_2$ data are converted to surface concentration by using a scaling algorithm. The zenith-sky surface $NO_2$ concentration data are assessed by comparison with in situ measurements. Lastly, in Section 5, several aspects of this zenith-sky surface $NO_2$ dataset are discussed, which include: diurnal and seasonal variation, and PBL effect,
followed by conclusions in Section 6.

## 2 Datasets and models

### 2.1 Measurements

#### 2.1.1 Pandora direct-sun total column NO$_2$

The Pandora instrument records spectra between 280 and 530 nm with resolution of 0.6 nm (Herman et al., 2009, 2015; Tzortziou et al., 2012). It uses a temperature-stabilized Czerny-Turner spectrometer, with a 50 µm entrance slit, 1200 groove mm$^{-1}$ grating, and a 2048 × 64 back-thinned Hamamatsu charge-coupled device (CCD) detector. The spectra are analysed using a total optical absorption spectroscopy (TOAS) technique (Cede, 2019), in which absorption cross sections for multiple atmospheric absorbers, such as ozone, NO$_2$, and sulphur dioxide (SO$_2$), are fitted to the spectra.

The Pandora direct-sun total column NO$_2$ data are produced using Pandora's standard NO$_2$ algorithm implemented in the BlickP software (Cede, 2019). The measured direct-sun spectra from 400 to 440 nm are used in the TOAS analysis. A synthetic reference spectrum is produced by averaging multiple measured spectra and corrected for the estimated total optical depth included in it. Cross sections of NO$_2$ at an effective temperature of 254.5 K (Vandaele et al., 1998), ozone at an effective temperature of 225 K (Brion et al., 1993, 1998; Daumont et al., 1992), and a fourth-order polynomial are all fitted. The resulting NO$_2$ SCDs are then converted to total column VCDs by using direct-sun geometry AMFs. Herman et al. (2009) show that Pandora direct-sun total column NO$_2$ has a clear-sky precision of 0.01 DU (in slant column) and a nominal accuracy of 0.1 DU (in vertical column, 2-sigma level). Additional information on Pandora calibrations, operation, and retrieval algorithms can be found in Herman et al. (2009) and Cede (2019).

The Pandora no. 103 instrument has been deployed in Toronto since September 2013 to perform direct-sun observations (Zhao et al., 2016). The instrument is installed on the roof of the Environment and Climate Change Canada (ECCC) Downsview building (43.7810°N, -79.4680°W) in Toronto. The building is located in a suburban area with multiple roads nearby. Since 2015, the instrument has employed an alternating direct-sun and zenith-sky observation schedule, which consists of direct-sun measurements every 90 seconds and zenith-sky measurements every 30 minutes during the sunlit period. About two-and-a-half years (February 2015 to September 2017) of continuous alternating measurements are used in this study.

#### 2.1.2 Pandora zenith-sky total column NO$_2$

Retrieval of trace gases from Pandora's zenith-sky measurements is not included in the standard BlickP processing software (Cede, 2019). The Pandora zenith-sky spectra for this study are processed using the differential optical absorption spectroscopy (DOAS) technique (Noxon, 1975; Platt, 1994; Platt and Stutz, 2008; Solomon et al., 1987) with the QDOAS software (Danckaert et al., 2015). A single reference spectrum is used, which was obtained from a zenith-sky measurement at local noon from a day that had low total column NO$_2$. Following the NDACC recommendations (Van Roozendael and Hendrick, 2012), NO$_2$ differential slant column densities (dSCDs) are retrieved in the 425-490 nm window (to retrieve oxygen collision complex simultaneously). The oxygen collision complex (O$_2$)$_2$ (referred here as O$_4$), which is created by the collision of two oxygen molecules, has broadband absorptions from UV to near IR spectral ranges (Greenblatt et al., 1990; Platt and Stutz,

2008; Thalman and Volkamer, 2013). $O_4$ is widely used as a reference gas by many DOAS applications to infer cloud and aerosol properties (e.g., Gielen et al., 2014; Wagner et al., 2004, 2014, 2016; Wang et al., 2015; Zhao et al., 2019). Cross sections of $NO_2$ at an effective temperature of 254.5 K (Vandaele et al., 1998), ozone at an effective temperature of 223 K (Bogumil et al., 2003), $H_2O$ (Rothman et al., 2005), $O_4$ (Hermans et al., 2003), and Ring (Chance and Spurr, 1997) are all fitted; a fifth-order polynomial and a first-order linear offset are also included in the DOAS analysis.

The output of QDOAS is $NO_2$ dSCDs, which can be converted to total column $NO_2$ via the Langley plot method with the use of the NDACC $NO_2$ AMF look-up table (LUT) (Van Roozendael and Hendrick, 2012). The NDACC AMF LUT is used here only as a reference since it was primarily developed for retrieval of stratospheric $NO_2$. Other empirical zenith-sky $NO_2$ AMFs have been developed and are used to convert $NO_2$ dSCDs to total columns. Details about these two different AMFs are given in Section 3.1.

### 2.1.3 OMI SPv3 data

The Ozone Monitoring Instrument (OMI) is a Dutch-Finnish nadir-viewing UV-visible spectrometer aboard the National Aeronautics and Space Administration (NASA)'s Earth Observing System (EOS) Aura satellite that was launched in July 2004. The OMI instrument measures the solar radiation backscattered by the Earth's atmosphere and surface between 270 and 500 nm with resolution of 0.5 nm (Levelt et al., 2006, 2018). OMI has a $780 \times 576$ CCD detector that measures at 60 across-track positions simultaneously, and thus, does not require across-track scanning. Due to this approach, the spatial resolution of the CCD pixels varies significantly along the across-track direction: those pixels near the track centre have ground footprint of 13 km $\times$ 24 km (along-track $\times$ across-track), whereas those close to the track edge (e.g. view zenith angle = 56°) have a ground footprint roughly of 23 km $\times$ 126 km (de Graaf et al., 2016). Note that from 2012 onwards the smallest pixels (across-track positions) can no longer be used and are excluded from the analysis (known as the "row anomaly", i.e. Levelt et al., 2018). This means the "smallest" pixels available for an OMI comparison are larger than 13 km $\times$ 24 km.

The OMI $NO_2$ data used in this work are the NASA standard product (SP) (Bucsela et al., 2013; Wenig et al., 2008) version 3.0 Level 2 (SPv3.0) (Krotkov et al., 2017). The $NO_2$ SCDs are derived using the DOAS technique in the 405-465 nm window (Marchenko et al., 2015). The AMFs used in SPv3.0 are calculated by using $1° \times 1.25°$ (latitude $\times$ longitude) resolution *a priori* $NO_2$ and temperature profiles from the Global Modeling Initiative (GMI) chemistry-transport model with yearly varying emissions (Krotkov et al., 2017).

### 2.1.4 In situ measurements

The National Air Pollution Surveillance (NAPS) network was established in 1969 to monitor and assess the quality of ambient (outdoor) air in the populated regions of Canada (https://www.canada.ca/en/environment-climate-change/services/air-pollution/monitoring-networks-data/national-air-pollution-program.html, accessed 23 November 2018). NAPS provides accurate long-term air quality data (ozone, $NO_2$, $SO_2$, carbon monoxide (CO), fine particulate matter, etc.) of a uniform standard across Canada(e.g., Dabek-Zlotorzynska et al., 2011; Reid and Aherne, 2016).

The in situ NO$_2$ data used in this study were collected at the NAPS Toronto North station (located 100 m away from the Pandora instrument). The site is 186 m above sea level, and the height of the air intake is 4 m above the ground.

The in situ NO$_2$ concentration is measured using a photolytic NO$_2$ instrument (Thermo 42i) that is also sensitive to other gaseous inorganic nitrogen compounds (e.g., nitric acid (HNO$_3$) and peroxyacetyl nitrate (PAN)) (McLinden et al., 2014).

Thus in areas where direct NO$_x$ (nitrogen oxides) emission sources are limited and other nitrogen compounds are present, NO$_2$ may be overestimated (e.g., in rural areas). For the current site, however, this positive bias has been found to be only about 5%, except for very low NO$_2$ concentrations (<5 ppbv) (Yushan Su, Ontario Ministry of the Environment, Conservation and Parks, personal communication, October 2018).

## 2.2 Numerical models

Predicted NO$_2$ fields from three atmospheric chemistry models are used in the algorithm described in Section 4.1 to derive surface NO$_2$ concentration from Pandora zenith-sky total column NO$_2$ data. Following McLinden et al. (2014), this work uses the Global Environmental Multi-scale Modelling Air quality and CHemistry (GEM-MACH) regional chemical transport model (CTM) and the GEOS-Chem global CTM to simulate total columns and vertical profiles of tropospheric NO$_2$ and surface NO$_2$ concentration. The stratospheric NO$_2$ partial columns are estimated using OMI satellite data and the Pratmo box-model.

**2.2.1 GEM-MACH**

GEM-MACH is ECCC's regional air quality forecast model. It is run operationally twice times per day to predict hourly surface pollutant concentrations over North America for the next 48 hours (Moran et al., 2009; Pavlovic et al., 2016; Pendlebury et al., 2018). The model consists of an online tropospheric chemistry module (Akingunola et al., 2018; Pavlovic et al., 2016) embedded within the ECCC Global Environmental Multi-scale (GEM) numerical weather prediction model (Côté

et al., 1998). Physical and chemical processes represented in GEM-MACH include emissions, dispersion, gas- and aqueous-phase chemistry, inorganic heterogeneous chemistry, aerosol dynamics, and wet and dry removal. The model uses gridded hourly emissions fields based on U.S. and Mexican national inventories from the U.S. Environmental Protection Agency (EPA) Air Emissions Modeling Platform and on Canada's national Air Pollutant Emission Inventory (APEI, https://pollution-waste.canada.ca/air-emission-inventory, accessed 23 November 2018) (Zhang et al., 2018). Currently, only NO$_x$ emissions in

the PBL are included in the operational model; free-tropospheric NO$_x$ emissions from lightning and in-flight aircraft are not considered. In this work, the GEM-MACH hourly NO$_2$ vertical profiles from 0 to 1.5 km and surface concentrations are retrieved from archived operational forecasts on the native model grid covering North America at 10 km $\times$ 10 km horizontal resolution for the period April 2016 to December 2017. The corresponding grid-box closest to the Pandora location was used in this study.

### 2.2.2 GEOS-Chem

The GEOS-Chem chemical transport model (Bey et al., 2001) has been used extensively in the retrieval of tropospheric columns, and has been shown to be capable of reasonably simulating the vertical distributions of $NO_2$ (Lamsal et al., 2008; Martin et al., 2002; McLinden et al., 2014). The model has a detailed representation of tropospheric chemistry, including

aerosols and their precursors (Park et al., 2004). In the simulation used in this study, a global lightning $NO_x$ source of 6 Tg N $yr^{-1}$ (Martin et al., 2002) was imposed. Lightning $NO_x$ emissions are computed as a function of cloud-top height, and are scaled globally as described by Sauvage et al. (2007) to match Optical Transient Detector/Lightning Imaging Sensor (OTD/LIS) climatological observations of lightning flashes. The model was run on a $1/2° \times 2/3°$ (latitude $\times$ longitude) grid in nested mode over North America, and was driven by assimilated meteorology from the Goddard Earth Observing System (GEOS-5). The

modelled $NO_2$ profiles were used to calculate monthly mean $NO_2$ partial columns in the free troposphere (1.5 to 12 km), as the GEM-MACH model does not include free-tropospheric $NO_2$ sources (lightning, in-flight aircraft emissions).

### 2.2.3 Pratmo box-model

Pratmo is a stratospheric photochemical box-model (Brohede et al., 2008; Lindenmaier et al., 2011; McLinden et al., 2000). The model has detailed stratospheric chemistry that includes long-lived species (nitrous oxide ($N_2O$), methane ($CH_4$), and

water vapor ($H_2O$)) and halogen families ($NO_y$, $Cl_y$, and $Br_y$) that are based on a combination of three-dimensional model output and tracer correlations (Adams et al., 2017). Heterogeneous chemistry of background stratospheric sulfate aerosols is also included. The model is constrained with climatological profiles of ozone and temperature.

Stratospheric $NO_2$ has a strong diurnal variation; therefore, diurnal corrections must be applied when OMI stratospheric $NO_2$ measurements (around local noon) are interpolated to Pandora measurement times. Ratios of modelled stratospheric $NO_2$

columns are calculated at OMI overpass time and Pandora measurement time. These ratios are multiplied by the OMI measured stratospheric $NO_2$ to produce stratospheric $NO_2$ columns corresponding to the time of Pandora measurements. Details about the use of the Pratmo box-model and the calculation of stratospheric $NO_2$ partial columns are provided in Section 4.1.

### 3 Total column $NO_2$ retrieval

### 3.1 Zenith-sky air mass factor

The NDACC UV-visible network uses zenith-sky AMFs in its total column $NO_2$ retrievals. To improve the overall homogeneity of the UV-visible $NO_2$ column measurements, NDACC recommended using the $NO_2$ AMF LUT (Van Roozendael and Hendrick, 2012). This LUT is based on climatological $NO_2$ profiles that are composed of (1) 20-60 km $NO_2$ profiles developed by Lambert et al. (1999, 2000) and (2) 12-20 km $NO_2$ profiles derived from SAOZ balloon observations (Van Roozendael and Hendrick, 2012). The $NO_2$ concentration is set to zero below 12 km altitude. The $NO_2$ AMFs have

been calculated using the UVSPEC/DISORT RTM (Hendrick et al., 2006; Wagner et al., 2007). The parameters used in

building the LUT are: wavelength, ground albedo, altitude of the station, and solar zenith angle (SZA). Aerosol extinction, ozone, and temperature profiles come from an aerosol model (Shettle, 1989), the U.S. Standard Atmosphere, and the TOMS V8 Climatology, respectively.

The NDACC LUT is designed for stratospheric $NO_2$ retrievals. Note that the absence of tropospheric $NO_2$ in the NDACC
LUT construction will lead to an underestimation of the total column $NO_2$ in urban areas. For example, from 2015 to 2017, tropospheric $NO_2$ accounted for $73 \pm 11$ % ($1\sigma$) of the total column amounts in Toronto (OMI SPv3.0 data). To account for this significant tropospheric $NO_2$ in urban areas, new empirical AMFs were developed in this study and the NDACC AMF LUT is used for comparison purposes only. In Tack et al. (2015), a more sophisticated four-step approach to derive total and tropospheric $NO_2$ columns from zenith-sky measurements was proposed, which involved with using a RTM to calculate
appropriate tropospheric AMFs. However, due to benefits from using the high-quality Pandora direct-sun total column $NO_2$ measurements, this work took a different but simple and robust approach to derive zenith-sky total column $NO_2$.

Empirical AMFs are calculated for Pandora zenith-sky $NO_2$ measurements in such a way that they can be used to retrieve zenith-sky total column $NO_2$ values that match the high-quality Pandora direct-sun total column $NO_2$ values. Inferring total columns from zenith-sky observations through comparisons with accurate direct-sun observations is a common approach for
Brewer and Dobson zenith-sky total ozone measurements (Kerr et al., 1988). For example, in the Brewer instrument zenith-sky ozone algorithm, weighted zenith-sky light intensities measured at four wavelengths ($F$) are expressed as a function of the slant path ($\mu$) and total column ozone (Kerr et al., 1981). The nine semi-empirical coefficients used to derive total column ozone from measured $F$ in the equation are estimated from a set of direct-sun and zenith-sky observations made nearly simultaneously (Fioletov et al., 2011). Instead of finding the link between zenith-sky spectral intensity and total
column values (i.e., following the Brewer and Dobson zenith-sky total ozone retrieval method), deriving empirical zenith-sky AMFs for Pandora zenith-sky measurements is more straightforward since Pandora zenith-sky spectra can be analyzed to produce $NO_2$ dSCDs.

The relation between VCD and dSCD can be expressed as:

$$VCD = \frac{dSCD + RCD}{AMF} \; , \qquad\qquad (1)$$

where, RCD is the reference column density that shows the slant column amount of the trace gas in the reference spectrum (Section 2.1.2). If we make an assumption that the coincident direct-sun (DS) and zenith-sky (ZS) measurements sampled the same air mass, then the empirical zenith-sky AMFs (referred to here as $AMF_{ZS\text{-}Emp}$) can be calculated by assuming $VCD_{DS} = VCD_{ZS}$, which gives

$$VCD_{DS}(SZA) = \frac{dSCD_{ZS}(SZA) + RCD_{ZS}}{AMF_{ZS-Emp}(SZA)} . \quad (2)$$

Next, we can use nearly-coincident $VCD_{DS}$ and $dSCD_{ZS}$ in a multi-non-linear regression to retrieve $AMF_{ZS\text{-}Emp}$ and $RCD_{ZS}$ together. To ensure the quality of the retrieved $AMF_{ZS\text{-}Emp}$, only high quality direct-sun total column $NO_2$ data are used with $SZA < 75°$. Details about the empirical zenith-sky AMF calculation are shown in Appendix A.

Figure 1 shows a comparison of the empirical zenith-sky AMFs and NDACC AMFs (calculated for the Toronto measurements). Total column $NO_2$ can then be retrieved using Eqn. (1) and these two sets of AMFs, where the one based on empirical AMFs is referred to as $VCD_{ZS-Emp}$ and the one based on NDACC AMFs is referred to as $VCD_{ZS-NDACC}$. The RCD value used in the retrievals is $0.39 \pm 0.01$ DU, which is retrieved along with $AMF_{ZS-Emp}$ (Appendix A). Figure 2 shows the comparisons of the $NO_2$ columns measured by zenith-sky and direct-sun methods. The regression analyses were performed by using the following coincidence criteria: (1) nearest Pandora direct-sun measurement that was within $\pm$ 5 min of Pandora zenith-sky measurement, (2) SZA < 75°, and (3) Pandora direct-sun total column $NO_2$ data have assured high quality (BlickP L2 data quality flag for nitrogen dioxide = 0). In general, the $VCD_{ZS-Emp}$ and $VCD_{ZS-NDACC}$ performed as expected. Compared with $VCD_{DS}$, the $VCD_{ZS-NDACC}$ shows a -25% bias, while the $VCD_{ZS-Emp}$ only shows a -4 % bias (indicated by the red lines on each panel and their slopes). In addition, $VCD_{ZS-Emp}$ shows less SZA dependence than $VCD_{ZS-NDACC}$ (see the increased bias for measurements made in larger SZA conditions in Figure 2b). These results confirm that, for urban sites, the tropospheric $NO_2$ profile should be included when calculating empirical zenith-sky AMFs. In the rest of the paper, only the zenith-sky $NO_2$ retrieved using empirical AMFs will be discussed. The derived zenith-sky total column $NO_2$ values are affected by both clouds and aerosols due to their impact on the light path. The presence of clouds and aerosols contributes to the uncertainty of the measurements. However, the impact of aerosols is expected to be moderate in most cases compare to that of clouds (e.g., Hendrick et al., 2011; Tack et al., 2015). Thus, this work has focused on evaluating the impact from clouds. Note that the Pandora zenith-sky total column $NO_2$ data discussed in Sections 3 are a "clear-sky subset" of Pandora zenith-sky measurements. The assessment of Pandora zenith-sky $NO_2$ measurements in cloudy conditions are provided in Section 4.

## 3.2 Comparison with satellite measurements

To illustrate the $NO_2$ variability over Toronto, Figure 3 shows the time series (2015-2017) from Pandora direct-sun, zenith-sky, and OMI SPv3.0 total column $NO_2$. In general, the $NO_2$ datasets from the ground-based Pandora instrument and the satellite follow the same pattern. However, the satellite data are likely to miss the peak $NO_2$ values in the morning since OMI only passes over Toronto once per day around 1:30 p.m. (local time).

We also performed regression analyses by using the following coincidence criteria: (1) nearest (in time) measurement that was within $\pm$ 30 min of OMI overpass time, (2) closest OMI ground pixel (having a distance from the ground pixel centre to the location of the Pandora instrument less than 20 km), and (3) cloud fraction <= 0.3 (the effective geometric cloud fraction, as determined by the OMCLDO2 algorithm; Celarier et al., 2016). In this comparison, only high-quality OMI data are used (VcdQualityFlags = 0) (Celarier et al., 2016). Figures 4a and 4b show the scatter plots of OMI vs. Pandora direct-sun and OMI vs. Pandora zenith-sky total column $NO_2$, respectively. Figures 4c and 4d show similar comparisons but only use OMI $NO_2$ measured by "small pixels" (i.e., having viewing zenith angle of less than 35°). The better correlation and lower bias for zenith-sky versus direct-sun might be a case of coincident errors, i.e., compare to Pandora direct-sun total column $NO_2$, both OMI and Pandora zenith-sky total column $NO_2$ underestimate the local $NO_2$ at Toronto (see Figure 2). When taking into account the standard error of the fitting and the confidence level of R, the difference between zenith-sky and direct-sun data is not

significant (i.e, in Fig. 4 from panels a to d, the slopes with standard error are $0.64 \pm 0.02$, $0.67 \pm 0.02$, $0.70 \pm 0.04$, and $0.71 \pm 0.03$; the 95% confidence intervals for R values are 0.45 to 0.63, 0.61 to 0.75, 0.43 to 0.77, and 0.60 to 0.86). The comparison results indicate that, at the Toronto site, OMI underestimates the total column by about 30 %. This underestimation is qualitatively consistent with the fact that the Pandora location is near the northern edge of peak Toronto $NO_2$, and the relatively large OMI pixels are also generally sampling areas of less $NO_2$ in the vicinity. The use of the relatively coarse (1°) GMI model for profiles shapes (Section 2.1.3) will also lead to a low bias considering the peak $NO_x$ emissions span roughly $0.5° \times 0.5°$. Similar results have been found elsewhere.

Ialongo et al. (2016) reported a similar negative bias using OMI SPv3.0 and Pandora direct-sun total column $NO_2$ in Helsinki (-32 % bias and R = 0.51), and they suggested this was due to the difference between the OMI pixel and the relatively small Pandora field-of-view. In Reed et al. (2015), Pandora measurements at 11 sites were evaluated; the authors found that the best correlation between OMI SPv3.0 and Pandora direct-sun total column $NO_2$ data is for rural sites. They concluded this could be due to smaller atmospheric variability in the rural region. Other studies such as Goldberg et al. (2017) found an even worse OMI-Pandora comparison between these two data products with striking negative bias at high values and poor correlation (R = 0.3). The authors attributed the poor agreement to the coarse resolution of OMI and its AMFs computed with GMI *a priori* $NO_2$ profiles. In general, our comparison results show that: (1) the Pandora direct-sun total column $NO_2$ data measured in Toronto have a reasonable agreement with OMI, and (2) the Pandora zenith-sky total column $NO_2$ data show results similar to those for direct-sun total column when compared with OMI SPv3.0.

## 4 Surface NO₂ concentration retrieval

The performance of the clear-sky Pandora zenith-sky total column $NO_2$ data has been assessed by using OMI and Pandora direct-sun data as described in Section 3.2. However, the validation of cloudy-scene Pandora zenith-sky total column data is not simple, since near-simultaneous good quality direct-sun or satellite measurements in most cloudy conditions are not available. This cloudy-scene validation can be done by comparison with in situ $NO_2$ measurements that are not affected by weather. In general, the comparison between total columns and surface concentrations can be done by two approaches: (1) convert Pandora zenith-sky total columns to surface concentrations; and (2) convert in situ surface concentrations to total column values. For example, Spinei et al. (2018) calculated "ground-up" VCDs from in situ surface concentrations by using additional measurements of PBL height or assuming trace gas profiles. In this work, the first approach is employed since the surface $NO_2$ data products from Pandora remote-sensing measurements have direct applications in areas such as air quality monitoring.

## 4.1 Column-to-surface conversion algorithm

A simple but robust scaling method is adapted to derive surface $NO_2$ concentration from Pandora zenith-sky total column $NO_2$ measurements. Following Lamsal et al. (2008) and McLinden et al. (2014), the surface $NO_2$ concentration is estimated using the modelled profile and surface concentration,

$$C_{pan} = \left(V_{pan} - V_{strat} - V_{ftrop}\right) \times \left(\frac{C}{V_{PBL}}\right)_{G-M}, \qquad (3)$$

where $C_{pan}$ is the surface $NO_2$ volume mixing ratio (VMR) to be estimated, $C$ is the surface $NO_2$ VMR from GEM-MACH (or G-M), $V_{pan}$ is the total column $NO_2$ measured by Pandora, $V_{strat}$ is the stratospheric $NO_2$ partial column, $V_{ftrop}$ is the $NO_2$ partial column in the free troposphere, and $V_{PBL}$ is the $NO_2$ partial column in the PBL. This equation assumes the chemical transport models can effectively capture the spatial and temporal behaviour of the concentration-to-partial-column ratio.

In this work, $V_{PBL}$ (0-1.5 km) is integrated from the GEM-MACH $NO_2$ profile and $V_{ftrop}$ (1.5-12 km) is integrated from the GEOS-Chem $NO_2$ profile. Both GEM-MACH and GEOS-Chem have an hourly temporal resolution. Thus, the integrated $V_{PBL}$ and $V_{ftrop}$ can account for $NO_2$ diurnal variation. However, $V_{strat}$ is from OMI monthly mean stratospheric $NO_2$, which does not have diurnal variation. Thus, the Pratmo box-model is used to calculate stratospheric $NO_2$ diurnal ratios. The OMI stratospheric $NO_2$ columns are interpolated to morning and evening hours by multiplying by the box-model diurnal ratios. Details about the calculation of $V_{strat}$ as well as references are provided in Appendix B.

The $(C/V_{PBL})_{G-M}$ ratio in Eqn. 3 is provided by GEM-MACH, and has hourly temporal resolution. This modelled $(C/V_{PBL})_{G-M}$ ratio is referred to here as a conversion ratio $R_{CV}$. Besides the hourly modelled conversion ratio, a simple monthly look-up table is built using an average of the one-and-a-half years of GEM-MACH model outputs (April 2016 to December 2017) that were available. The look-up table (referred to here as the Pandora surface-concentration look-up table, or PSC-LUT) is composed of monthly conversion ratios with hourly resolution as shown in Figure 5. For example, assuming that a Pandora $NO_2$ total column measurement is made on a day in December at 15:00 LST, then the corresponding conversion ratio from the PSC-LUT is 28 ppbv $DU^{-1}$ (see the black arrow). Our results in Figure 5 show that the conversion ratio changes throughout the day as well as with season: 0.1 DU (partial column $NO_2$ in the PBL) corresponds to 5-8 pptv of surface $NO_2$ in the morning (8:00 LST), 2-3 pptv around local noon (13:00 LST), and 2-4 pptv in the evening (18:00 LST). In general, the variation of conversion ratios demonstrates that the surface $NO_2$ concentration is controlled not only by PBL height, but also by both boundary-layer dynamics and photochemistry. The surface $NO_2$ derived using the hourly modelled $R_{CV}$ ratio is referred to here as $C_{pan-model}$, while the surface $NO_2$ derived using the monthly mean PSC-LUT is referred to here as $C_{pan-LUT}$. In general, $C_{pan-model}$ is a data product that depends on daily model outputs, but $C_{pan-LUT}$ only needs the pre-calculated PSC-LUT and is thus less dependent on the model. In general, the look-up table approach ($C_{pan-LUT}$) is aiming for a quick and near-real-time data delivery. Thus, to minimize year-to-year variation (e.g., from changing meteorological conditions or changing local emission patterns), for a given year we recommend using a mean PSC-LUT that is calculated from model simulations of previous years. On the other hand, the $C_{pan-model}$ is the off-line, high-quality, year-specific data product that will be delivered for air quality research and other applications. Details of these two different surface $NO_2$ data products are discussed in the next section.

## 4.2 Comparison with measurements and model

Figure 6 shows the evaluation of modelled and Pandora zenith-sky surface $NO_2$ concentrations, both using in situ $NO_2$ measurements as the reference. The Pandora data have been filtered for heavy clouds (details are given in Section 4.3). The GEM-MACH modelled surface concentrations in Toronto reproduce the in situ measurements very well with the comparison showing high correlation (R = 0.78) and moderate positive bias (37 %, Figure 6a). The Pandora zenith-sky surface $NO_2$ data, $C_{pan-model}$, shows almost the same correlation (R = 0.77), with only -7 % bias (Figure 6b). The better performance of $C_{pan-model}$ is expected since the conversion method for Pandora zenith-sky measurements relies on the GEM-MACH modelled $NO_2$ profile (see Eqn. 3); in other words, the Pandora zenith-sky surface $NO_2$ has at least one more piece of information (i.e., $NO_2$ total column) than GEM-MACH surface $NO_2$ concentrations. The $C_{pan-LUT}$ shows a similar correlation coefficient (R = 0.73) and has improved bias (-3 %, Figure 6c). This result (slightly lower correlation) is also reasonable and acceptable since $C_{pan-LUT}$ is derived with the monthly PSC-LUT, which has less accurate information than the hourly modelled data.

Besides the improved bias, Pandora zenith-sky surface $NO_2$ concentrations, $C_{pan-model}$ and $C_{pan-LUT}$ (Figures 6e and 6f) also have better frequency distributions than the GEM-MACH (Figure 6d). Figure 6d shows that the $NO_2$ surface concentrations peaks (ambient background concentrations) from model and in situ data are misaligned. This indicates that the GEM-MACH $NO_2$ background surface concentrations have a 1ppbv low bias at this site. In contrast, the zenith-sky surface $NO_2$ at peak-frequency matches the in situ data (Figures 6e and 6f), indicating that the low bias of the background surface $NO_2$ value has been corrected with this additional information from Pandora zenith-sky total column measurements. In addition, in high $NO_2$ concentration conditions (> 20 ppbv), the zenith-sky surface $NO_2$ also shows better agreement with the in situ $NO_2$ than do the modelled data. The mean of the top 10 % of the in situ data is 26 ± 1 ppbv (uncertainty of the mean), whereas the corresponding values for GEM-MACH, $C_{pan-model}$, and $C_{pan-LUT}$ are 39 ± 1 ppbv, 26 ± 1 ppbv, and 27 ± 1 ppbv, respectively.

The total column-to-surface concentration conversion algorithm has also been applied to the Pandora direct-sun total column $NO_2$ (see Figure 7). Figure 7b shows that the direct-sun surface $NO_2$ data have a similar agreement with the in situ data (-8 % bias and R = 0.80) as the zenith-sky surface $NO_2$. In high $NO_2$ concentration conditions, direct-sun data have a similarly good agreement with the in situ measurements. For this direct-sun based dataset, the mean of the top 10 % of the in situ data is 27 ± 1 ppbv, whereas the corresponding values for GEM-MACH, $C_{pan-model}$, and $C_{pan-LUT}$ are 40 ± 1 ppbv, 27 ± 1 ppbv, and 27 ± 1 ppbv, respectively

Thus, in general, both Pandora zenith-sky and direct-sun surface $NO_2$ datasets can be used reliably to obtain surface concentrations. The good consistency between $C_{pan-model}$, and $C_{pan-LUT}$ implies that two versions of Pandora surface $NO_2$ data can be delivered in the future, i.e., an off-line version that relies on the inputs from hourly model, and a near-real-time version that only needs a pre-calculated LUT.

## 4.3 Measurements in different sky conditions

Although zenith-sky observations are less sensitive to cloud conditions than direct-sun observations, we still need to be cautious about the derived zenith-sky surface $NO_2$ in heavy cloud conditions. Due to enhanced scattering, heavy clouds could lead to a significant overestimation of surface $NO_2$ derived from zenith-sky measurements. A cloud filtering method based on retrieved $O_4$ dSCDs is used to identify these conditions. High retrieved $O_4$ values correspond to long optical path lengths and therefore it is expected that corresponding $NO_2$ values are overestimated as discussed in Appendix C.

The effectiveness of the zenith-sky $NO_2$ in cloudy scenes is demonstrated by the time series plots (Figure 8) of in situ and Pandora direct-sun and zenith-sky data (in their original temporal resolutions). Under clear-sky conditions (for example, April 8-14), both Pandora direct-sun and zenith-sky-based surface concentrations correlate well with the in situ measurements. Under moderately cloudy conditions, when Pandora direct-sun observations cannot provide high-quality data, Pandora zenith-sky observation still can yield good measurements that compare well with in situ data (for example, April 26-29). Under heavy cloud conditions, however, which are identified by enhanced $O_4$ (Appendix C), Pandora zenith-sky-derived surface $NO_2$ yielded higher than in situ measurements (for example, April 4 and 6, see the green squares). This feature is due to the enhanced multi-scattering in heavy cloud conditions, which leads to enhanced $NO_2$ absorption in the measured spectra.

Sensitivity tests (Appendix C) show that only 10 % of all zenith-sky measurements are strongly affected by this enhanced absorption, indicating the zenith-sky $NO_2$ algorithm is applicable to most measurements made in thin and moderate cloud conditions (Toronto has about 44 % of daylight hours with clear-sky conditions per year). The relative strength of direct-sun measured by a collocated Total Sky Imager (model TSI-880) is plotted on top of each panel in Figure 8 as an additional indicator of sky conditions. The relative strength of direct-sun is from the integration of blocking-strip luminance. In general, when the relative strength of direct-sun is high (> 60), good quality direct-sun and zenith-sky $NO_2$ data can both be produced. However, when sun strength is moderate (30-60), only zenith-sky $NO_2$ data are reliable. When sun strength is low (< 30), zenith-sky $NO_2$ has increased bias and needs to be filtered out.

## 5 Discussion

This study evaluated the performance of Pandora zenith-sky measurements with Pandora direct-sun measurements, satellite measurements, and in situ measurements. In general, the quality of zenith-sky data is affected by three main factors: (1) quality of empirical zenith-sky AMFs; (2) cloud conditions (heavy clouds or moderate/thin clouds); and (3) quality of modelled $NO_2$ profile (this factor only applies to Pandora surface $NO_2$ data). The quality of empirical zenith-sky AMFs and the cloud effect have been addressed in Appendices A and C, respectively. The third factor is discussed in Sections 5.1 and 5.2. The uncertainty estimations for Pandora zenith-sky and direct-sun data products are provided in Appendix D.

## 5.1 Diurnal and seasonal variation

From the Pandora zenith-sky and direct-sun measurements, and modelled $NO_2$ profiles, surface $NO_2$ concentrations were obtained that agree well with in situ measurements collected at the same location. The Pandora surface $NO_2$ data were also analyzed in more detail with a focus on temporal variations. Figure 9 shows the averaged surface $NO_2$ diurnal variations of four different datasets. The in situ instrument produces continuous measurements 24 hours per day, whereas Pandora only has measurements when sunlight is available. The diurnal variation of surface $NO_2$ concentration is controlled by dynamics (e.g., vertical mixing, wind direction), photochemistry, and local emissions. Thus, the diurnal variations are calculated using only the hours when in situ, direct-sun, and zenith-sky data are all available.

Figure 9 shows that all four datasets/curves captured the enhanced morning surface $NO_2$ and the decreasing trend afterwards. However, the model has a positive offset (6-9 ppbv) in the morning (due in part to the use of older emissions inventories: Moran et al., 2018) and a negative offset (1-3 ppbv) in the evening relative to the in situ data. For example, at 7:00 LST, in situ $NO_2$ is $14.9 \pm 9.3$ ppbv, while GEM-MACH, Pandora DS, and Pandra ZS $NO_2$ are $23.5 \pm 15.0$ ppbv, $15.6 \pm 10.5$ ppbv, and $15.2 \pm 6.8$ ppbv, respectively. At 17:00 LST, in situ $NO_2$ is $7.3 \pm 5.8$ ppbv, while GEM-MACH, Pandora DS, and Pandora ZS $NO_2$ are $5.6 \pm 5.0$ ppbv, $3.6 \pm 2.6$ ppbv, and $5.2 \pm 3.4$ ppbv, respectively. The larger standard deviations in the morning are due to the datasets not being divided into work-days and weekends. Compared to the modelled data, the Pandora direct-sun and zenith-sky data show improvements in the morning, but almost no changes for the evening. This feature is investigated and found to be correlated with the GEM-MACH modelled PBL height (details in Section 5.2).

The diurnal variation is also examined by grouping the data by seasons. Figure 10 shows that the surface $NO_2$ concentrations in winter (December, January, and February) are higher than the corresponding values in summer (June, July, and August). This difference is mainly due to short sunlit periods and less solar radiation (e.g., increased lifetime of $NO_2$ and decreased PBL height) in winter. The model has better agreement with the in situ data in summer than in the colder seasons. The best performance of the model is found around local noon, and this feature is not dependent on seasons. Figure 10 also shows that the quality of Pandora zenith-sky and direct-sun surface $NO_2$ estimates is affected by the quality of GEM-MACH modelled data. For example, Figure 10c shows that in autumn (September, October, and November), GEM-MACH has the largest offset in the morning. This error is thus propagated to the Pandora direct-sun surface data, and leads to a larger offset in the morning (than any other season). On the other hand, when GEM-MACH shows a better agreement with in situ measurements (e.g., in spring and summer), Pandora zenith-sky and direct-sun estimates also show better agreement with in situ observations. In general, both Pandora direct-sun and zenith-sky surface $NO_2$ data show good agreement with in situ measurements in all seasons; the hourly mean values of Pandora surface $NO_2$ are all well within the 1σ envelope of the in situ measurements.

## 5.2 Planetary boundary-layer effect

The larger morning offset in modelled surface $NO_2$ may indicate that the GEM-MACH modelled PBL heights are biased in the morning when the boundary layer is shallow. Figure 11 (left column) shows the modelled PBL height plotted as a function

of the difference between modelled and in situ surface $NO_2$. Figure 11a shows that, in general, the difference between modelled and in situ $NO_2$ decreases with an increase of PBL height. When the modelled PBL height is less than 100 m, the mean difference is $18 \pm 12$ ppbv (1σ), while when the modelled PBL height is 1 km, the mean difference is only $2.9 \pm 6.4$ ppbv. Even though the modelled surface concentrations are significantly impacted by the PBL, the modelled conversion ratio (from column to surface concentrations) seems unaffected since the surface $NO_2$ concentrations derived from Pandora zenith-sky data ($C_{pan-model}$) show much less dependence on the PBL height (Figure 11b). When the modelled PBL height is less than 100 m, the mean difference is $0.9 \pm 8.9$ ppbv. When the modelled PBL height is 1 km, the mean difference is slightly improved to $0.1 \pm 4.4$ ppbv. Figures 11c and 11h show similar plots as Figure 11a and 11b, but the dataset has been divided into three time-bins (before 9:00, 11:00 to 13:59, and after 15:00). Figures 11c, 11e, and 11f confirm that whenever the modelled PBL height is low, the relative difference between the model and in situ data is high. However, in general, most of these shallow PBL height conditions occur in the morning, and thus the modelled surface $NO_2$ has larger bias compared to in situ data in the morning. Figures 11d, 11f, and 11h show that Pandora zenith-sky surface $NO_2$ data have similar performance for all these three time-bins, which indicates that the data have less PBL height dependency than the modelled data. In other words, the model is able to capture the ratio between the boundary layer partial column and surface $NO_2$, although the PBL height may not be correct in the model. When this ratio is applied to both Pandora direct-sun and zenith-sky data, the estimated surface concentrations agree better with the in situ measurements.

## 6 Conclusions

The Pandora spectrometer was originally designed to retrieve total columns of trace gases such as ozone and $NO_2$ from direct-sun spectral measurements in the UV-visible spectrum. In this work, a new zenith-sky total column $NO_2$ retrieval algorithm has been developed. The algorithm is based on empirical AMFs derived from nearly simultaneous direct-sun and zenith-sky measurements. It is demonstrated that this algorithm can retrieve total columns in thin and moderate cloud conditions when direct-sun measurements are not available: only 10 % of the measurements affected by heavy cloud have to be filtered out due to large systematic biases (68 %). The new Pandora zenith-sky total column $NO_2$ data shows only -4% bias compared to the standard Pandora direct-sun data product. In addition, OMI $NO_2$ SPv3.0 data demonstrate similar biases (-30 % and -29 %, respectively) when compared to direct-sun and zenith-sky Pandora total column $NO_2$ data.

Surface $NO_2$ concentrations were calculated from Pandora direct-sun and zenith-sky total column $NO_2$ using column-to-surface ratios derived from GEM-MACH regional chemical transport model. The bias between Pandora-based direct-sun and zenith-sky $NO_2$ surface concentration estimates and in situ measurements is only -8 % and -7 % (with correlation coefficients 0.80 and 0.77), respectively, while the bias between the modelled concentrations and in situ measurements is up to 37 %. The Pandora-based surface $NO_2$ concentrations also show good diurnal and seasonal variation when compared to the in situ data. High surface $NO_2$ concentrations in the morning (from 6:00 to 9:00, local standard time) are present in all measured and modelled datasets, while, on average, the model overestimates surface $NO_2$ in the morning by 8.6 ppbv (at 7:00 LST). It

appears that the bias in modelled surface $NO_2$ is related at least in part to an incorrectly diagnosed PBL height. In contrast, the difference between Pandora-based and in situ $NO_2$ does not show any significant dependence on the PBL height. Thus, to enable a fast and practical Pandora surface $NO_2$ data production, the use of a pre-calculated conversion ratio PSC-LUT is recommended.

5    The new retrieval algorithm for Pandora zenith-sky $NO_2$ measurements can provide high-quality $NO_2$ data (both total column and surface concentration) not only in clear-sky conditions, but also in thin and moderate cloud conditions, when direct-sun observations are not available. Long-term Pandora zenith-sky $NO_2$ data could be used in future satellite validation for the medium cloudy scenes. Moreover, a column-to-surface conversion look-up table was produced for the Pandora instruments deployed in Toronto; therefore, quick and practical Pandora-based surface $NO_2$ concentration data can be obtained for air

10   quality monitoring purposes. The variation of conversion ratios in the PSC-LUT demonstrates that the surface $NO_2$ concentration is controlled not only by the PBL height, but also by both boundary-layer dynamics and photochemistry. This conversion approach can also be used to derive surface concentrations from satellite VCD measurements and thus can be particularly useful for the new generation of geostationary satellite instruments for air quality monitoring such as the Tropospheric Emissions: Monitoring of Pollution (TEMPO, Zoogman et al., 2014). Currently, the standard Pandora

15   observation schedule includes direct-sun, zenith-sky, and multi-axis scanning measurements (i.e., measuring at multiple viewing angles). At present, multi-axis measurement algorithms are still under development, but in the future, by using the multi-axis measurements and optimal estimation techniques (e.g., Rodgers, 2000) or the five angles O2O2-ratio algorithm (Cede, 2019), it may be possible for Pandora measurements to be used to derive $NO_2$ tropospheric profiles and columns.

*Data availability.* Pandora data are available from the Pandonia network (http://pandonia.net/data/). In situ surface $NO_2$ data are available from the National Air Pollution Surveillance (NAPS) program (http://maps-cartes.ec.gc.ca/rnspa-naps/data.aspx). OMI $NO_2$ SPv3.0 data are available from https://disc.gsfc.nasa.gov/. Any additional data may be obtained from Xiaoyi Zhao (xiaoyi.zhao@canada.ca).

*Author contributions.* XZ analyzed the data and prepared the manuscript, with significant conceptual input from DG, VF, and CM, and critical feedbacks from all co-authors. JD, AO, VF, XZ, and SCL operated and managed the Canadian Pandora network. AL, MDM, and DG performed and analyzed the GEM-MACH simulations. AC, MT, and MM operated the Pandonia network and provided critical technical support to the Canadian Pandora measurements and subsequent data analysis.

*Acknowledgements.* Xiaoyi Zhao was supported by the NSERC Visiting Fellowships in Canadian Government Laboratories program. We thank Ihab Abboud and Reno Sit for their technical support of Pandora measurements. We thank NAPS for providing surface $NO_2$ data. We acknowledge the NASA Earth Science Division for providing OMI $NO_2$ SPv3.0 data. We also thank Thomas Danckaert, Caroline Fayt, Michel Van Roozendael, and others from IASB-BIRA for providing the QDOAS

software, the NDACC UV-visible working group for providing NDACC UV-visible $NO_2$ AMF LUT, and Yushan Su from the Ontario Ministry of the Environment, Conservation and Parks for providing NAPS Toronto North station in situ $NO_2$ information. We thank two anonymous referees for their helpful and insightful comments, which improved the overall quality of this work.

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

## Appendices

### A. Empirical zenith-sky AMF

Before calculating the empirical zenith-sky AMF, the $VCD_{DS}$ and $dSCD_{ZS}$ have both been strictly filtered to ensure any measurements used in this calculation have the highest quality. For $VCD_{DS}$, data are filtered following Cede (2019) with several factors being considered, such as wavelength shift and residual in spectra fitting, direct-sun AMF, and estimated uncertainties for the vertical column. For $dSCD_{ZS}$, data are filtered using similar criteria as for $VCD_{DS}$, with adjustments for zenith-sky observations.

The $VCD_{DS}$ and $dSCD_{ZS}$ data are merged and divided into several SZA bins. Each bin covers 5°. A multi-non-linear regression is performed by using the following equation:

$$\begin{bmatrix} VCD_1 \\ \vdots \\ VCD_n \end{bmatrix} = \begin{bmatrix} dSCD_1 & \cdots & 0 \\ \vdots & \ddots & \vdots \\ 0 & \cdots & dSCD_n \end{bmatrix} \begin{bmatrix} b_1 \\ \vdots \\ b_n \end{bmatrix} + RCD \begin{bmatrix} I_1 & \cdots & 0 \\ \vdots & \ddots & \vdots \\ 0 & \cdots & I_n \end{bmatrix} \begin{bmatrix} b_1 \\ \vdots \\ b_n \end{bmatrix} \quad (4)$$

where, $VCD_n$ is not a single direct-sun VCD data point, but is an m × 1 matrix (m is the total number of measurements in SZA bin number $n$); the $VCD_n$ represents all direct-sun VCDs in a 5° SZA bin, and each element of the m × 1 matrix is a single VCD in that SZA bin. Similarly, $dSCD_n$ is also an m × 1 matrix, with each element representing a single coincident zenith-sky dSCD in SZA bin number $n$. $I_n$ is an m × 1 indicator function, where the elements of $I_n$ are set to 1. The $RCD$ and $b_1$ to $b_n$ are the parameters to be retrieved. In short, the design of this regression is based on Eqn. 2 (Section 3.1). The idea is to retrieve zenith-sky AMFs in several SZA bins, and, at the same time, all these regressions in different SZA bins are constrained to share a common predictor (RCD). The regression model can be solved by using an iterative procedure (Seber and Wild, 2003) to yield the estimated coefficients, $b_1$ to $b_n$ and RCD. The $b_n$ is the reciprocal of zenith-sky AMF in SZA bin $n$.

This regression model has been evaluated by using different sizes for the SZA bins. A 5° SZA bin is selected because the SZA bin must be small enough to capture the SZA dependency on zenith-sky AMFs, and, at the same time, it must also be large enough to ensure a sufficient number of measurements in each SZA bin (to perform reliable regressions). In order to deal with

the diurnal variation of $NO_2$ concentration and changing of profile shape (e.g., due to changing of boundary layer heights), the dataset has been divided into morning and evening sets, and discrete AMFs are retrieved for a.m. and p.m. separately (see the blue and red squares with error bars in Figure 1).

Next, these discrete AMF values are used to fit an empirical zenith-sky $NO_2$ AMF function, which has the expression:

$$AMF = a_1 + (1.02 - a_1)/cos(SZA) \qquad (5).$$

The fitted empirical zenith-sky AMFs are shown in Figure 1 as blue and red lines (data regression period from February 2015 to September 2017). Several sensitivity tests have been performed to assess the quality of the empirical zenith-sky AMFs, including fitting the AMFs with/without a diurnal difference, fitting the AMFs with different empirical functions (e.g., exponential and simple geometry approximation) and fitting the AMFs by seasons. All these different choices of empirical AMFs fitting functions or methods only introduce less than 5 % difference in the retrieved empirical AMFs. Thus, to make the empirical AMFs simple and robust, we selected to fit with a diurnal difference (Eqn. 5). In addition, the current empirical AMFs are limited to high and intermediate sun conditions (i.e., SZA< 75°). For low-sun conditions, the total AMF for zenith-sky measurements is expected to be a strong function of not only the SZA, but also the tropospheric column itself. Thus, for future work to derive low-sun empirical zenith-sky AMFs, the stronger influence of PBL $NO_2$ has to be accounted (i.e., the geometry form AMFs are not enough).

### B. Stratospheric NO₂ column

Several stratospheric $NO_2$ column values were tested and used in the surface $NO_2$ concentration algorithm (Eqn. 3). Figure A1a shows the OMI monthly mean (referred to as OMI) and Pratmo box-model stratospheric column $NO_2$ (Adams et al., 2016; McLinden et al., 2000) (referred to as box). Since the satellite only samples Toronto once per day, the OMI stratospheric $NO_2$ lacks diurnal variation. To account for the diurnal variation, diurnal ratios of $NO_2$ VCD have been calculated and applied to OMI monthly mean data. The stratospheric $NO_2$ columns are calculated using

$$V_{OMI}(t) = \frac{V_{box}(t)}{V_{box}(t_0)} \times V_{OMI}(t_0) , \qquad (6)$$

where, $V_{OMI}(t_0)$ is the OMI measured stratospheric column, $t_0$ is OMI overpass time, $V_{box}(t_0)$ is the modelled stratospheric column at OMI overpass time, $V_{box}(t)$ is the modelled stratospheric column at time $t$, and $V_{OMI}(t)$ is the interpolated stratospheric column at time $t$. The interpolated OMI stratospheric columns are referred to as OMI-box. The grey dots on Figure A1b are OMI-box stratospheric $NO_2$ columns. The monthly mean of the box model (blue line) and OMI-box (black line) show that the amplitude of OMI-box is larger than the amplitude of the box model.

To justify why this diurnal variation has to be included, Figure A1c shows the total column $NO_2$ time series. The diurnal stratospheric $NO_2$ variation is about 0.1 DU in the summer (see grey dots in Figure A1b) when Pandora measured monthly mean total column is about 0.5 DU (Figure A1c). Thus, neglecting this diurnal variation will lead to diurnal biases in the derived surface $NO_2$ data (e.g., in the morning, this will lead to the overestimation of the stratospheric $NO_2$ and thus the underestimation of surface $NO_2$). Please note that the strength of this bias is related to 1) the $NO_2$ profile (weights between

stratospheric and tropospheric $NO_2$), and 2) the observation geometry (direct-sun or zenith-sky). In general, an urban site with direct-sun observation should have smaller impact from the stratospheric diurnal variation. On the other hand, a rural site with zenith-sky observation should have a significant impact.

**C. Cloud effect and heavy cloud filtration**

Direct-sun measurements need an unobscured sun. Even thin clouds could decrease the quality of retrieved $NO_2$ total columns, especially for low altitude clouds. Unlike direct-sun measurements, zenith-sky observations are made with scattered sunlight and have limited sensitivity to cloud cover. For example, Hendrick et al. (2011) calculated that, for NDACC UV-visible zenith-sky ozone measurements, clouds only contribute 3.3 % to the total random error. This is because a trace gas that is mostly distributed in the stratosphere has the mean scattering layer located at a higher altitude than the cloud layer. However, this

assumption may not be valid for $NO_2$. Depending on the properties of the clouds and the $NO_2$ profile, the clouds could have non-negligible impacts on zenith-sky $NO_2$ observations.

A typical method of removing zenith-sky measurements affected by heavy clouds is to eliminate measurements with large enhancements of $O_4$ and/or $H_2O$ (Van Roozendael and Hendrick, 2012). In the Pandora zenith-sky $NO_2$ retrieval, we use the $O_4$ dSCDs. Since the measured $O_4$ dSCDs has SZA dependency, all measured $O_4$ dSCDs are plotted against SZA and a second

order quantile regression (Koenker and Hallock, 2001) is applied to select the top few percentile of the measured $O_4$ dSCDs. Figure A2 shows examples of selected Pandora zenith-sky $NO_2$ data and their corresponding $O_4$ dSCDs values. For example, Figure A2a shows the $O_4$ dSCDs versus SZA and the top 10 percentile of the data with enhanced $O_4$ are marked in grey. The corresponding Pandora zenith-sky data are plotted against in situ data in Figure A2b, which shows low correlation (R = 0.34) and high bias (68 %). This result indicates that the enhanced scattering due to heavy cloud caused a positive bias in the Pandora

zenith-sky $NO_2$ retrieval. Figures A2c and A2d are similar to Figures A2a and A2b, but for selected Pandora zenith-sky $NO_2$ data that have $O_4$ values within the $40^{th}$ to $50^{th}$ percentile range. Figure A2d shows that when $O_4$ is not enhanced, the derived zenith-sky $NO_2$ has good agreement with in situ data (R = 0.8 and bias = -5 %). To summarize how the retrieved $O_4$ dSCDs can indicate the quality of the Pandora zenith-sky $NO_2$, the results from the other percentile bins are shown in Figure A3. In general, besides the top 10 percentile of data, the results from all the other bins show good correlation (above 0.6) and low

bias. Thus, in this study, the Pandora zenith-sky $NO_2$ data that have $O_4$ values within only the top 10 percentile are considered to be affected by heavy clouds and are removed. Some examples of this heavy cloud effect are shown in Figure A4 and Figure 8 in Section 4.3.

**D. Uncertainty estimation**

The uncertainties of retrieved Pandora zenith-sky $NO_2$ data products (total column and surface concentration) are estimated

and discussed here to assess the quality of the data products. The uncertainties of total column and surface concentrations are estimated first using the uncertainty propagation method (referred to here as the UP method) based on Eqns. 2 and 3. The combined uncertainties of total column can be calculated using:

$$\sigma_{VCD_{ZS}} = \sqrt[2]{\left(\frac{\sigma_{dSCD}}{AMF}\right)^2 + \left(\frac{\sigma_{RCD}}{AMF}\right)^2 + \left(\frac{\sigma_{AMF \times SCD}}{AMF^2}\right)^2},\qquad\qquad (7)$$

where $\sigma_{dSCD}$ is the statistical uncertainty on the DOAS fit (output of QDOAS) and $\sigma_{RCD}$ and $\sigma_{AMF}$ are the estimated statistical uncertainties using standard errors of the RCD and the zenith-sky empirical AMF regression, respectively (Eqn. 4). To estimate the upper limit of the nominal uncertainty, AMF and SCD are used as median and maximum values in the dataset, respectively.

The combined uncertainties of the surface concentration can be calculated using:

$$\sigma_{C_{Pan}} = \sqrt[2]{\left(R_{CV}\sigma_{V_{Pan}}\right)^2 + \left(R_{CV}\sigma_{V_{strat}}\right)^2 + \left(R_{CV}\sigma_{V_{ftrop}}\right)^2 + \left(V_{Pan} - V_{strat} - V_{ftrop}\right)^2 \sigma_R^2},\qquad (8)$$

where $\sigma_{Vpan}$ is the uncertainty of Pandora zenith-sky total column $NO_2$, (here we use the derived $\sigma_{VCD}$ in Eqn. 7), $\sigma_{Vstrat}$ is the uncertainty of the stratospheric $NO_2$ column (estimated using the 1-sigma standard deviation of the $V_{strat}$), $\sigma_{Vftrop}$ is the uncertainty of the free troposphere $NO_2$ column (estimated using the 1-sigma standard deviation of the $V_{ftrop}$). $R_{CV}$ is the GEM-

MACH calculated surface VMR to PBL column ratio, and $\sigma_R$ is the uncertainty of that ratio (estimated using the 1-sigma standard deviation of the $R_{CV}$). The means of $R_{CV}$, $V_{Pan}$, $V_{strat}$, and $V_{ftrop}$ are used in the uncertainty estimation.

Besides the UP method, another simple approach to estimate uncertainty is to compare the data product with another high-quality (lower uncertainty) coincident data. For example, if we assume that the Pandora direct-sun total column $NO_2$ data can represent the true value, we can estimate the uncertainty of Pandora zenith-sky total column $NO_2$ by calculating the 1-sigma

standard deviation of their difference (referred to here as the SDD method):

$$\sigma_{VCD_{ZS}} = \sigma(VCD_{DS} - VCD_{ZS}).\qquad\qquad (9)$$

Similarly, if we assume that the in situ surface $NO_2$ VMR can represent the true value, the uncertainty of Pandora zenith-sky-based surface $NO_2$ VMR can be given by:

$$\sigma_{C_{Pan}} = \sigma(C_{insitu} - C_{pan}).\qquad\qquad (10)$$

Also, if there is systematic bias between the two datasets, it can be removed and the random uncertainty can be calculated by:

$$\sigma_{VCD_{ZS}} = \sigma(VCD_{DS} - k_1 VCD_{ZS}),\qquad\qquad (11)$$

$$\sigma_{C_{Pan}} = \sigma(C_{insitu} - k_2 C_{pan}),\qquad\qquad (12)$$

where $k_1$ and $k_2$ are the slopes in the linear fits with intercept set to zero (e.g., slopes in Figs. 2 and 6). This method is referred to here as the unbiased SDD. These three uncertainty estimation methods (UP, SDD, and unbiased SDD) were all implemented,

and the results are summarized in Table A1. The results show that Pandora zenith-sky total column $NO_2$ data have a 0.09-0.12 DU uncertainty that is about twice to the Pandora direct-sun total column nominal accuracy (0.05 DU, at 1-sigma level). When using the UP method, for the worst-case scenario, the Pandora zenith-sky total column $NO_2$ have a 0.17 DU uncertainty (i.e. using minimum of AMFs to estimate the upper limit of uncertainty). The estimated Pandora zenith-sky-based surface $NO_2$ VMR data have uncertainties from 4.8 to 6.5 ppbv. In Eqn. 8, the contributions of the $V_{Pan}$, $V_{Strat}$, $V_{ftrop}$, and $R_{CV}$ terms to the

total uncertainty are 36%, 2%, 0.3%, and 62 %, respectively. This result indicates that the uncertainty in the Pandora zenith-sky-based surface $NO_2$ VMR is dominated by the uncertainties of Pandora zenith-sky total column $NO_2$ and the modelled column-to-surface conversion ratio ($R_{CV}$). However, note that this uncertainty budget depends on the $NO_2$ vertical distributions,

and hence may vary from site to site; e.g., in Toronto, tropospheric column $NO_2$ is typically 2-4 times higher than stratospheric column $NO_2$, and thus, the contribution to uncertainty from $V_{Pan}$ is much larger than the corresponding contributions from $V_{Strat}$ and $V_{ftrop}$. In addition, the uncertainty of Pandora direct-sun surface $NO_2$ VMR is also estimated and provided in Table 1. It shows slightly better results than for zenith-sky-based surface $NO_2$ VMR.

**Table 1. Estimated uncertainties for Pandora zenith-sky total column and surface NO₂.**

| Estimation method | $\sigma_{VCD_{ZS}}$ (DU) | $\sigma_{C_{Pan-ZS}}$ (ppbv) | $\sigma_{C_{Pan-DS}}$ (ppbv) |
|:---:|:---:|:---:|:---:|
| UP | 0.12 | 6.5 | 5.4 |
| SDD | 0.09 | 5.1 | 5.0 |
| unbiased SDD | 0.09 | 4.8 | 4.8 |

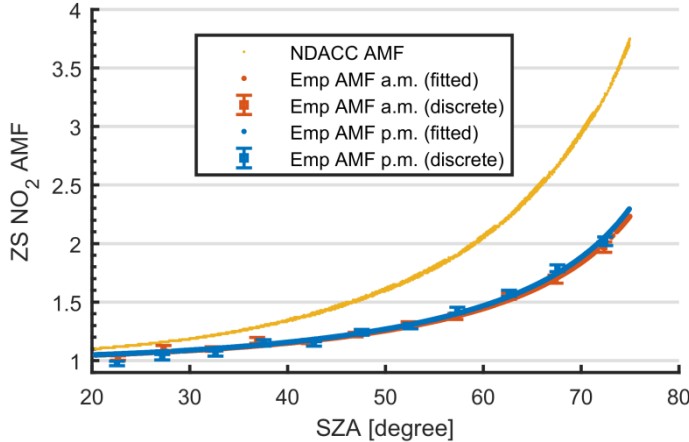

**Figure 1. Comparison of zenith-sky NO₂ air mass factors. Blue and red squares with error bars (standard error) represent the empirical discrete zenith-sky NO₂ AMFs in each SZA bin for Toronto for the period Feb. 2015 to Sept. 2017. Blue and red lines show the fitted empirical zenith-sky NO₂ AMFs. NDACC AMFs calculated using the NDACC look-up table and assuming no NO₂ in the troposphere are shown in yellow.**

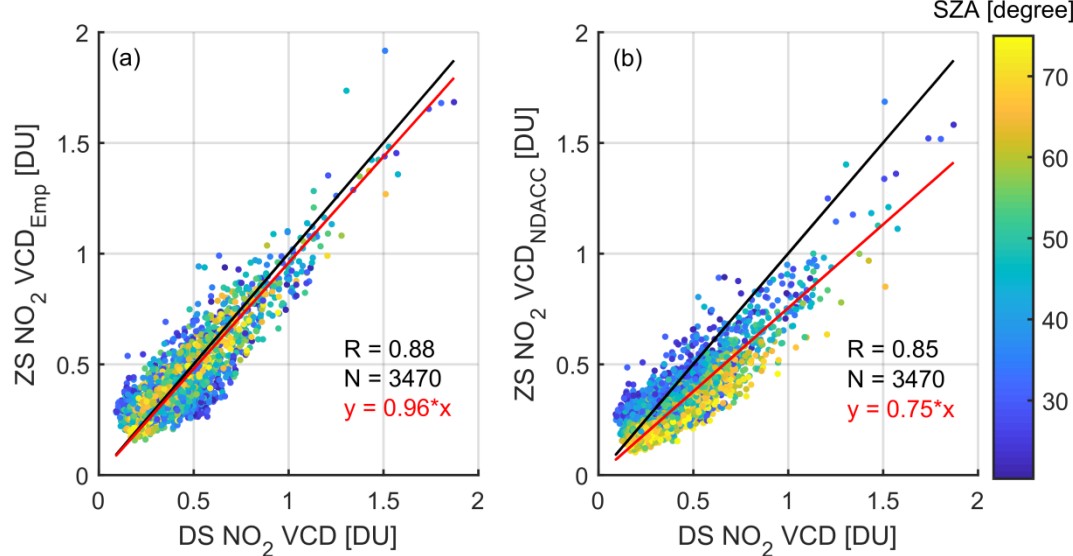

**Figure 2. Comparisons of NO₂ total columns (2015-2017): (a) zenith-sky total column NO₂ retrieved using empirical AMFs vs. direct-sun total column NO₂, (b) zenith-sky total column NO₂ retrieved using NDACC AMFs vs. direct-sun total column NO₂. On each scatter plot, the red line is the linear fit with intercept set to 0, and the black line is the one-to-one line. The scatter plot is colour-coded by solar zenith angle (SZA).**

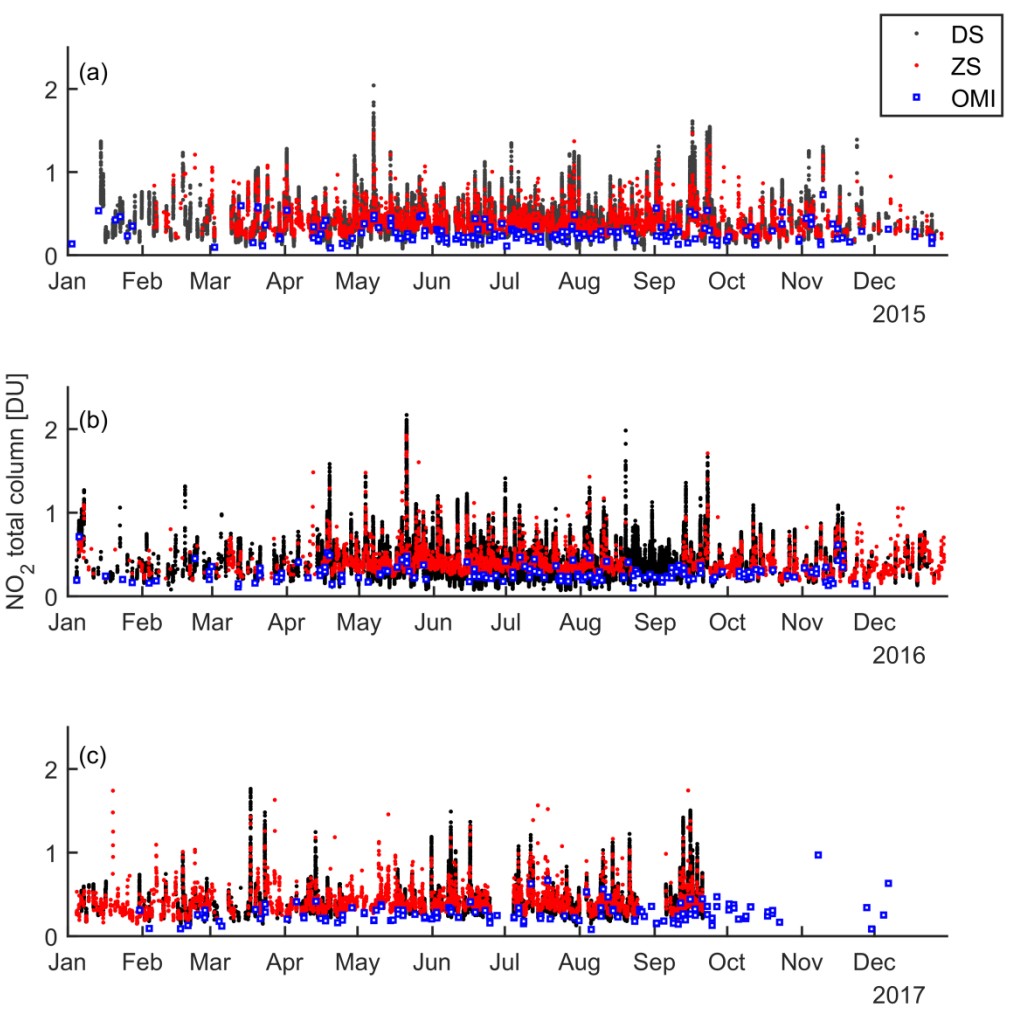

**Figure 3. Annual time series of Pandora direct-sun (DS), Pandora zenith-sky (ZS), and OMI SPv3 total column NO₂ in Toronto from 2015-2017.**

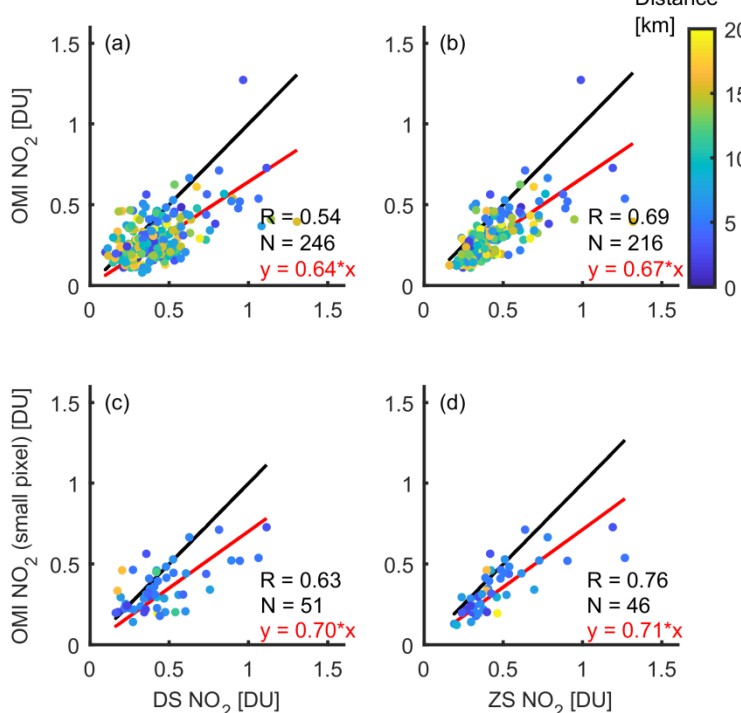

**Figure 4.** OMI vs. Pandora total column NO$_2$ (2015-2017). (a) and (c) show OMI vs. Pandora direct-sun NO$_2$, and (b) and (d) show OMI vs. Pandora zenith-sky NO$_2$. (a) and (b) show all available OMI measurements, while (c) and (d) show OMI data from small pixels only. On each scatter plot, the red line is the linear fit with intercept set to 0 and the black line is the one-to-one line. All scatter plots are colour-coded by the distance from the centre of an OMI ground-pixel to the location of Pandora.

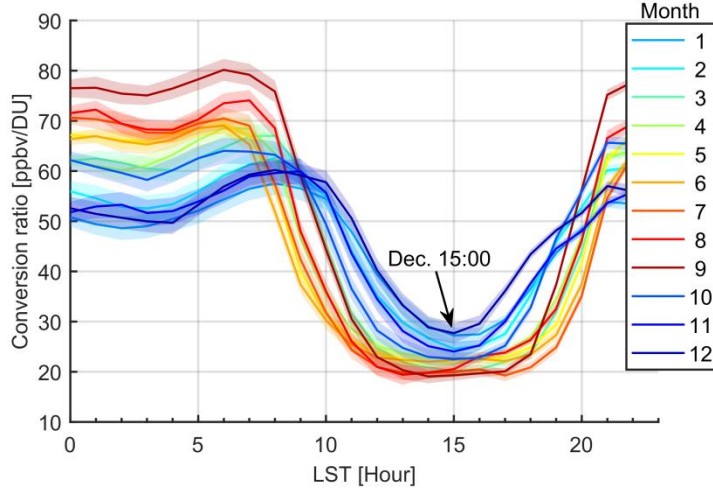

**Figure 5. Dependence of the Pandora surface NO$_2$ concentration look-up table (PSC-LUT) on month of year and hour of day. The PSC-LUT is constructed using the GEM-MACH modelled NO$_2$ conversion ratios. Solid lines are monthly mean conversion ratios colour-coded by month. The shaded envelopes are the standard error of the mean.**

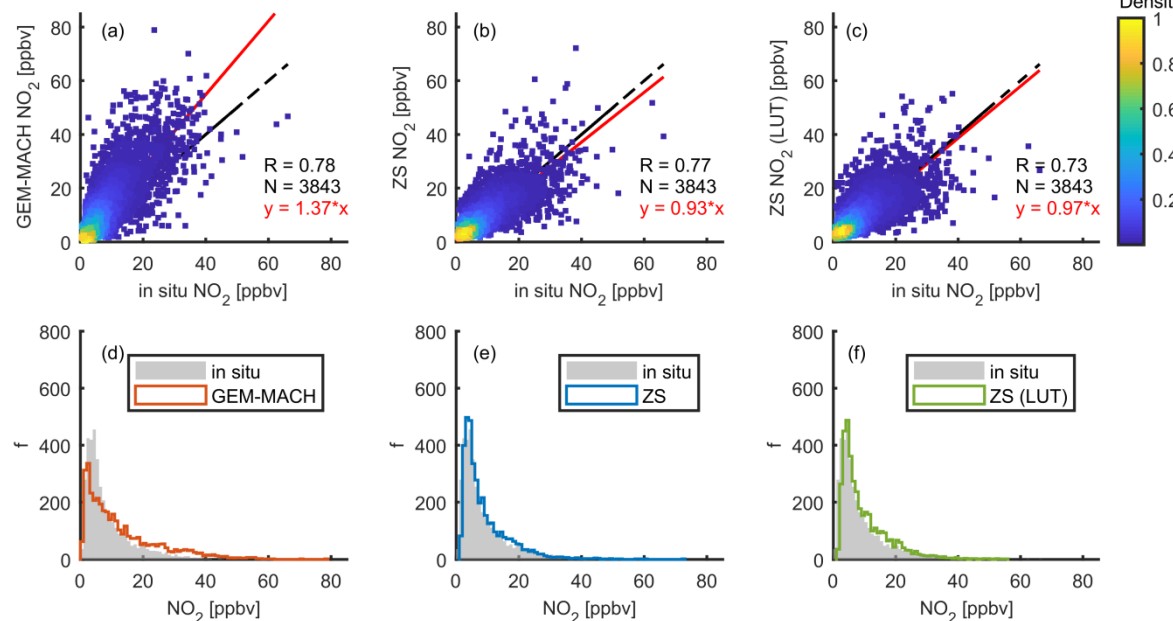

**Figure 6. Modelled and Pandora zenith-sky surface NO₂ vs. in situ NO₂ (2016-2017). (a) shows the GEM-MACH modelled surface NO₂ data vs. in situ NO₂; (b) and (c) show the Pandora zenith-sky (ZS) surface NO₂ data vs. in situ NO₂. The Pandora ZS surface NO₂ data in (b) and (c) are derived using the hourly modelled conversion ratio and the monthly PSC-LUT, respectively. (d) to (f) are histograms corresponding to the data in (a) to (c). On each scatter plot, the red line is the linear fit with intercept set to 0 and the black line is the one-to-one line. The scatter plots are colour-coded by the normalized density of the points.**

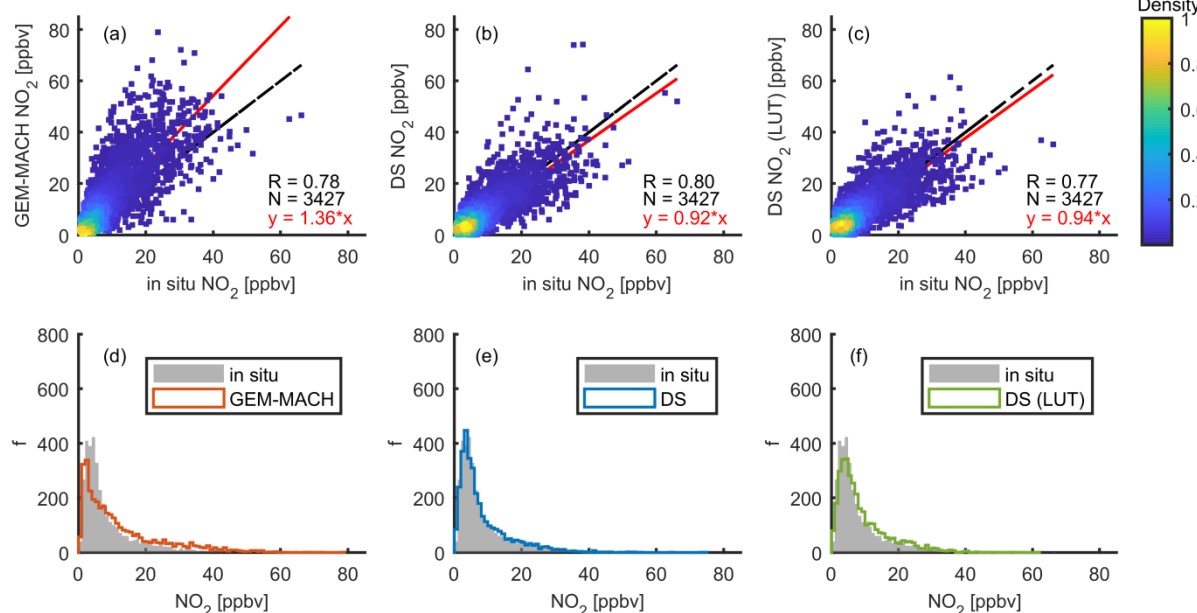

**Figure 7. Modelled and Pandora direct-sun surface NO₂ vs. in situ NO₂ (2016-2017). (a) shows the GEM-MACH modelled surface NO₂ data vs. in situ NO₂; (b) and (c) show the Pandora direct-sun (DS) surface NO₂ data vs. in situ NO₂. The Pandora DS surface NO₂ data in (b) and (c) are derived using the hourly modelled conversion ratio and the monthly PSC-LUT, respectively. (d) to (f) are histograms corresponding to the data in (a) to (c). On each scatter plot, the red line is the linear fit with intercept set to 0 and the black line is the one-to-one line. The scatter plots are colour-coded by the normalized density of the points.**

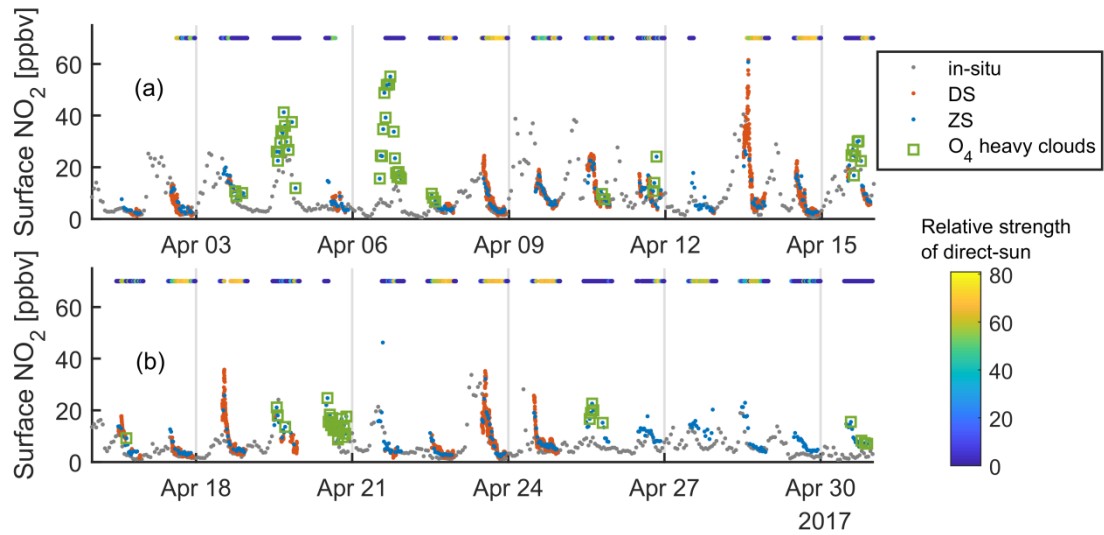

**Figure 8. Example of surface NO₂ concentration time series in all conditions (April 2017). The in situ, Pandora direct-sun (DS), and Pandora zenith-sky (ZS) surface NO₂ concentrations are shown by different coloured dots. The total sky imager relative strength of direct-sun data are plotted as a colour-coded horizontal dot-line on the top area of each panel. For Pandora zenith-sky data, the measurements with enhanced O₄ (heavy cloud indicator) are also labelled by green squares.**

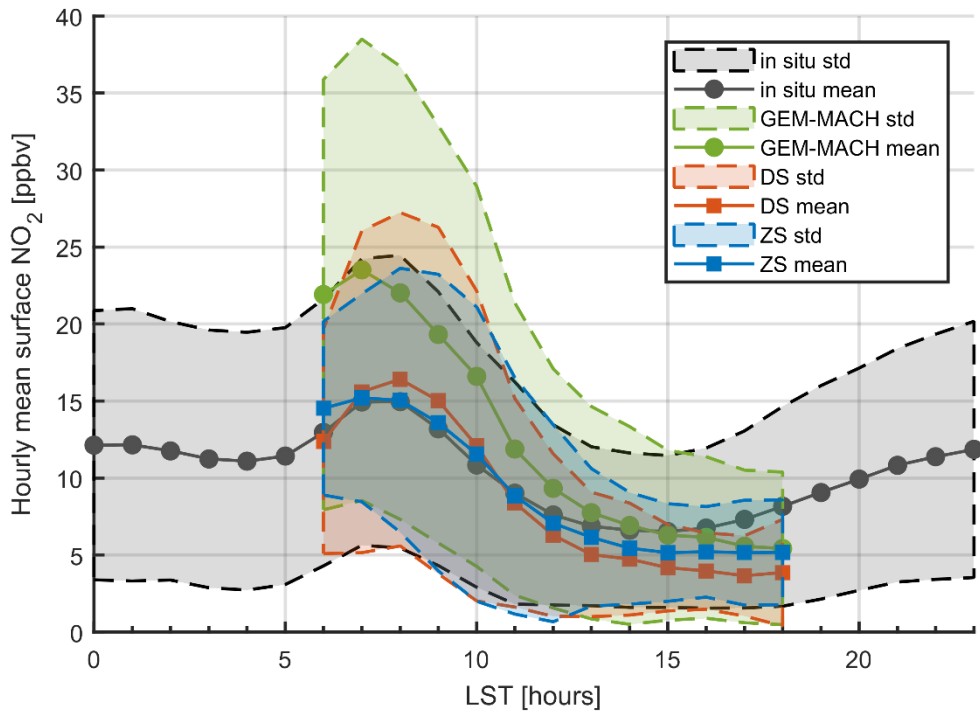

**Figure 9. Diurnal variation of surface NO₂ concentration (2016-2017). The x-axis is the local standard time (LST). Lines with dot/square symbols represent the hourly mean of corresponding data indicated by the legend. The shaded area represents the 1σ envelope.**

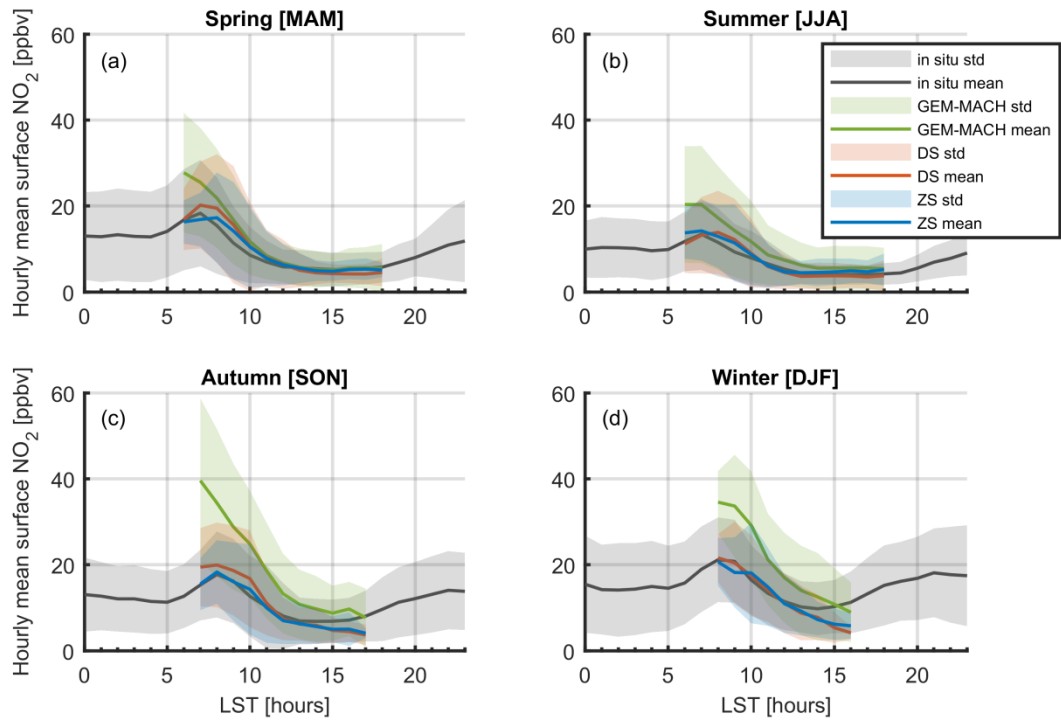

**Figure 10. Diurnal variation of surface NO₂ concentration by season (2016-2017). The x-axis is the local standard time (LST). Each panel represents data collected in one season (spring, summer, autumn or winter). Solid lines represent mean of corresponding data indicated by the legend. The shaded area represents the 1σ envelope.**

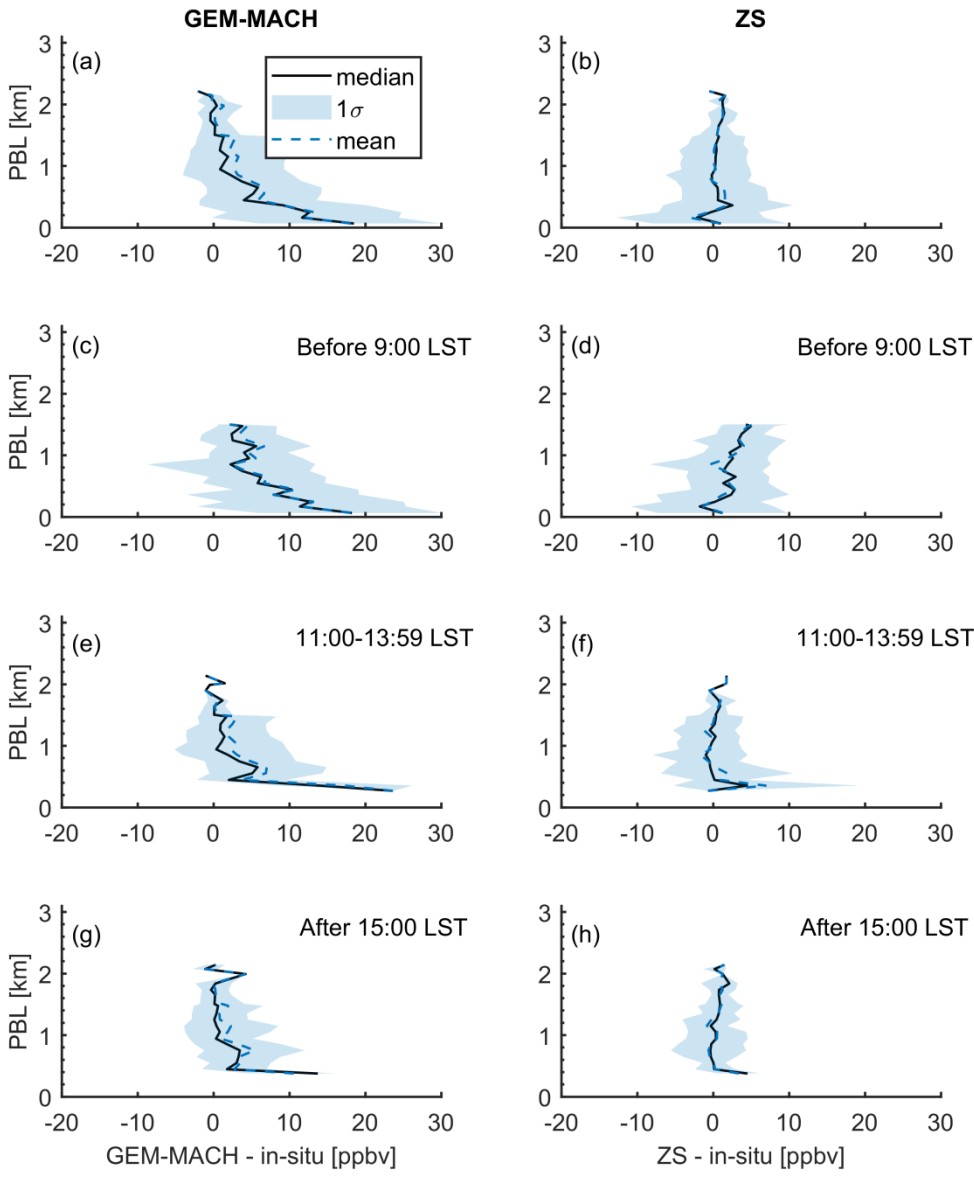

**Figure 11. Illustration of planetary boundary-layer (PBL) effect (2016-2017). The y-axis is planetary boundary-layer height in km. The x-axes for the left column are the difference between GEM-MACH and in situ surface NO₂ concentrations; the x-axes for the right column are the difference between Pandora zenith-sky ($C_{pan\text{-}model}$) and in situ surface NO₂ concentration. Panels a and b show all available data, panels c and d show the morning data (before 9:00, local standard time), panels e and f show the noon data (from 11:00 to 13:59), and panels g and h show the evening data (after 15:00).**

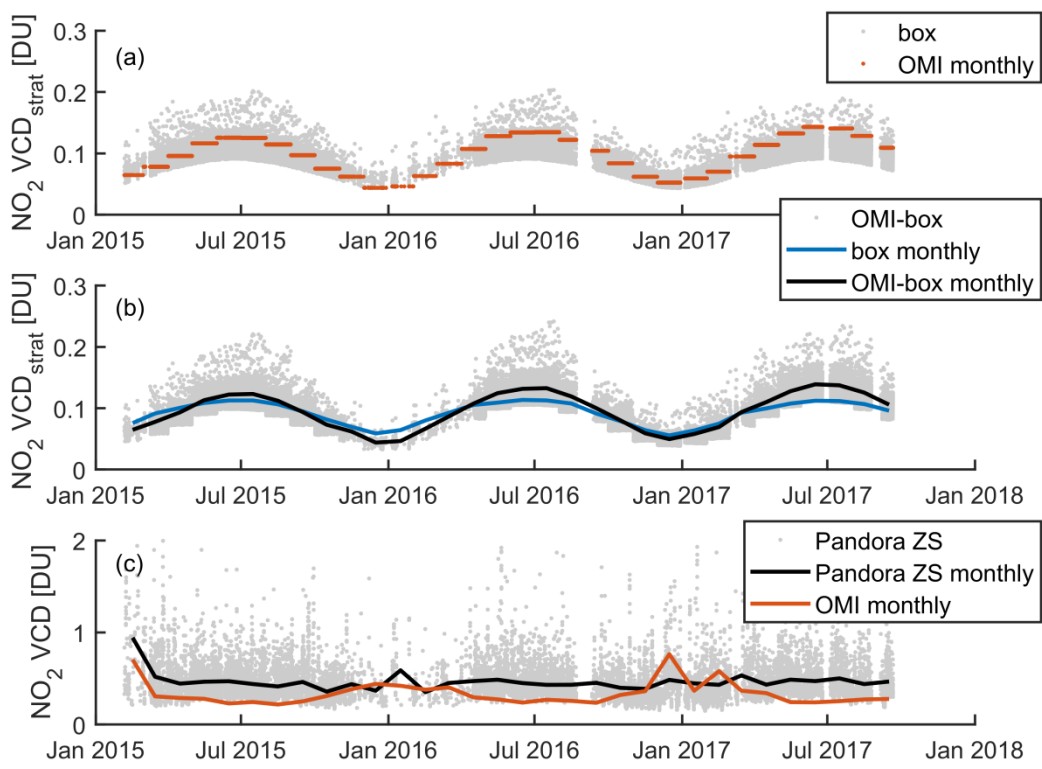

**Figure A1.** Time series of measured and modelled NO₂ columns: **(a)** stratospheric columns from the box model (hourly) and OMI (monthly), **(b)** stratospheric columns from OMI-box (hourly), box (monthly) and OMI-box (monthly), and **(c)** total columns from Pandora zenith-sky and OMI.

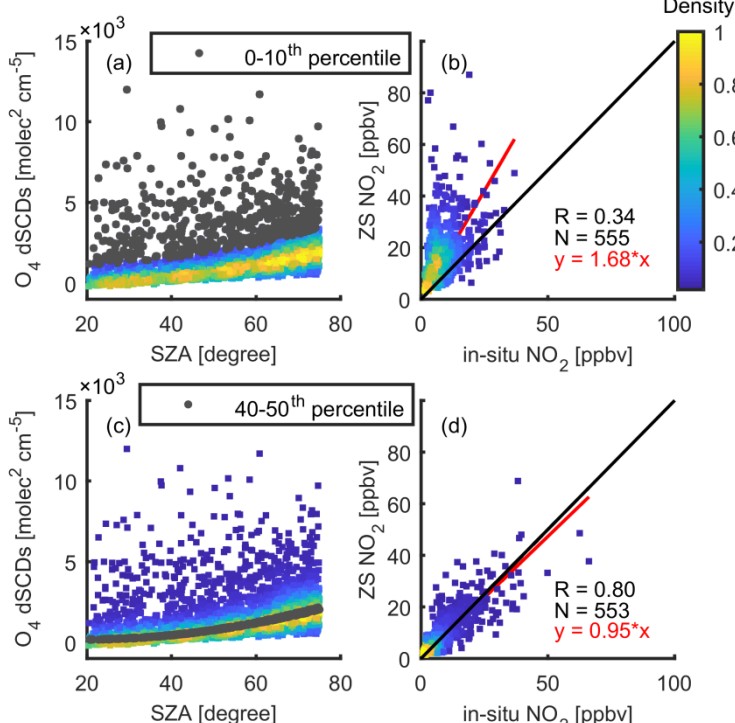

**Figure A2. Illustration of cloud effect and heavy-cloud data filtration: (a) shows measured O₄ differential slant column densities versus solar zenith angle, the gray dots represent the top 0-10th percentile range of O₄, (b) shows the scatter plot of zenith-sky versus in situ surface NO₂ use data that has O₄ value within 0-10th percentile range (as identified in (a)), (c) is similar to (a) but the gray dots represent the 40-50th percentile range of O₄, (d) is similar to (b) but use the data that has O₄ value within the 40-50th percentile range. On scatter plot (b) and (c), the blue line is the linear fit with intercept set to 0, the red line is a simple linear fit, and the black line is the one-to-one line. All plots are colour-coded by the normalized density of the points.**

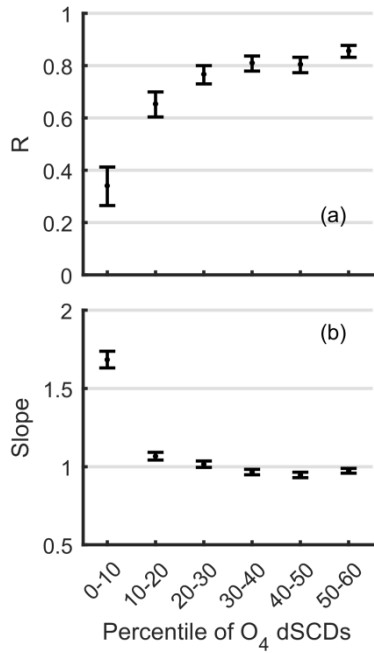

**Figure A3. Correlation coefficient and bias (slope) between zenith-sky and in situ surface NO₂ data in different O₄ dSCDs percentile bins. (a) shows the correlation coefficients, (b) shows the slopes of linear fit with intercept set to 0.**

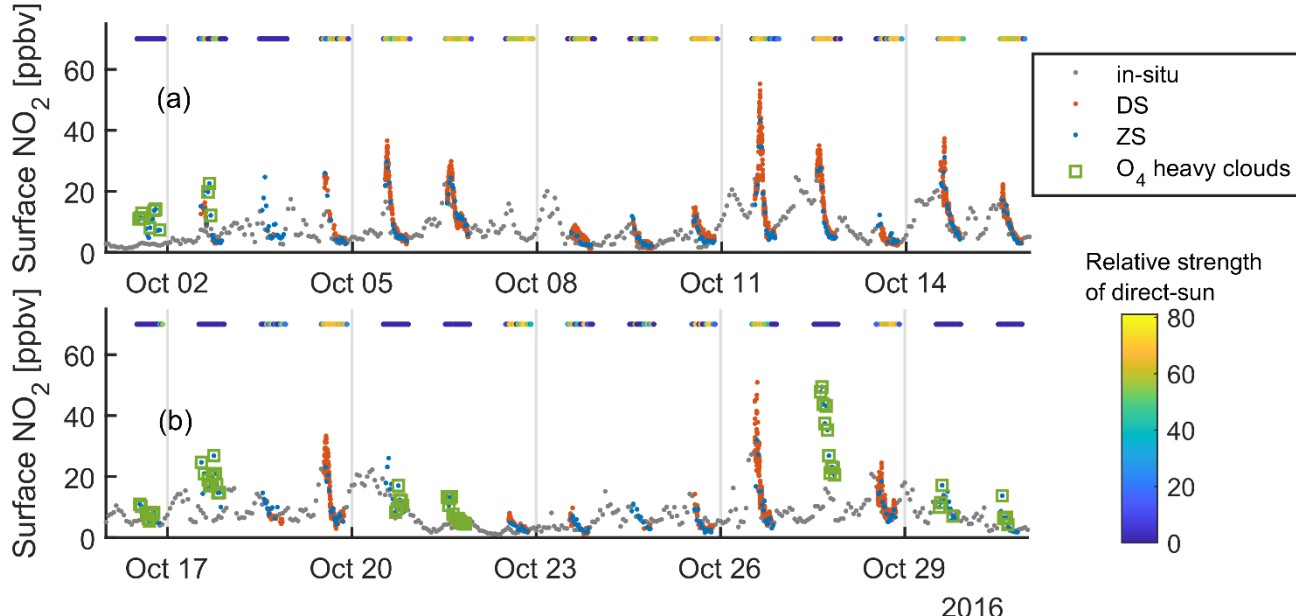

**Figure A4. Example of surface NO₂ concentration time series in all conditions. The in situ, Pandora direct-sun (DS), and Pandora zenith-sky (ZS) surface NO₂ concentrations are shown by different coloured dots. The total sky imager relative strength of direct-sun data are plotted as a colour-coded horizontal dot-line on the top area of each panel. For Pandora zenith-sky data, the measurements with enhanced O₄ (heavy cloud indicator) are also labelled by green squares.**