# Peer review of "Retrieval of total column and surface NO2 from Pandora zenith-sky measurements"

_Atmospheric Chemistry and Physics, 2018_

## Referee Comment (RC1) · Anonymous Referee #1 · 4 Mar 2019

**General comments**

This paper describes the process to obtain the surface NO2 mixing ratio using the NO2 column from the ground-based Pandora observation. Some validations are performed by the comparison to the model outputs and in-situ measurement. The approaches look logical and the final products seem reasonable. Considering the rising importance of Pandora data for the air quality monitoring, publication of this work is useful to the research community and potential readers. But some more discussions should be included to improve this manuscript. Please refer to the comments below for the revision process.

Specific comments

P1, L27: Traffic is one of the important NO2 sources. Please include.

P2 L1-L12: Why is the total vertical column NO2 significantly treated first instead of insitu measurement. For air quality monitoring, in-situ measurement has been performed basically. Before the importance of Pandora observation is addressed, the necessity of ground-based remote sensing of trace gases should be stated first, in spite of the well-structured ground-based in-situ networks.

P2 L8-9: The reason to use zenith-sky observations under the cloudy condition is not clear. Why zenith-sky mode has more reliable than direct-sun mode under the cloudy condition?

P4 L20-21: Why do we need O4 retrieval? Still, some potential readers do not have any idea about the meaning of O4 retrieval.

P7 L1-2: In other Pandora paper (usually including Dr. Jay Herman in the author list), diurnal variation of stratospheric NO2 is not much considered. It will be more interesting (and even useful) to add some arguments compared with their approaches.

P8 L10-24: Notation is not clear. There are VCD\_DS and VCE\_ZS. Also, there are VCD\_EMP and VCD\_NDACC. VCD\_EMP (and VCD\_NDACC) is related to VCD\_DS or VCD\_ZS? Better notation or more description of these notations look required.

P10 L10-12: V\_ftrop is from the GEOS-chem simulations. How does GEOS-Chem consider the lightning NOx in the free-troposphere? In other words, GEOS-Chem can catch the free-tropospheric lightening NOx reasonably? It seems better to include some discussions about the estimation of lightning NOx in the free troposphere and even stratosphere, which is related to the final quality of C\_pan from the Pandora observation.

P10 L22-24: How about the year-to-year variation of conversion ratio? Is it larger or smaller than the diurnal variation of conversion ratio? If there is, it seems to include some discussion for how to treat or consider the year-to-year variation.

ACPD
P11 L21-28: How to connect the results and discussions here to those in Fig. 4? The lesson from Fig. 4 seems that the ZS Pandora NO2 has better quality than the direct-sun Pandora NO2, right? But here both ZS and direct-sun Pandora NO2 shows consistent quality. Then what do we really learn from Fig. 4 in this work?

P12 L3-18 and Fig. 8: In this study, the validation is performed for the 2015-2017 period. Why only a month (APR 2017) is considered in this analysis? The better quality of ZS NO2 than direct-sun NO2 can be justified with this figure but whether the ZS NO2 always provide the reliable NO2 under the heavy clouds cannot be determined based on only 1-month situation. It seems necessary to deal with additional cloudy cases (at least in the supplements).

P13 L3-L20 and Figs. 9 and 10: In addition to the GEM-MACH model results, Pandora ZS and direct-sun NO2 products also show the higher standard deviation after sunrise compared to the time close to sunset. Considering the radiative extinction is more disturbed during the time close to sunrise and sunset, the higher standard deviation in the morning is not well understood. Why NO2 product in the morning has higher biases? This is also due to the propagation of GEM-MACH PBL uncertainty? (But authors mentioned that the surface NO2 from ZS Pandora observation is less dependent on the PBL height in P13 L29-30).

**Technical corrections**

P6 L2: two times -> twice

P7 L30-32: The meaning of this statement is not clear to me. Please more clarify the explanation.

P8 L10-12: This statement looks redundant, just a repetition of statements above.

P11 L10: This result (slightly lower correlation) -> This slightly lower correlation

Figs. 4, 7, and A2: If a slope value is the only significant value, then 'slope = \*\*\*' looks better expression. If authors would show the information with the format of 'y = slope
x', it seems better to add intercept values together, to have the original regression equation (the equation looks not completed without intercept values)

Figure captions: It seems that all figure captions can be more clarified to better describe the figures. Please check and correct once more before the submission of revised manuscript (Now they are somewhat confused. Fig. 6 and 7 are examples. What really figure b and c imply? Direct description of figure a, b, c looks better to read).

Time period: Please clearly suggest the time period of each analysis (at least in each figure caption). I think each analysis has a different time period of analysis, which is a little complex to me. While I can accept this difference, the clear time information should be provided to help the readers' understanding.

---

## Referee Comment (RC2) · Anonymous Referee #2 · 25 Apr 2019

**General comments**

This study presents a new approach to extend direct-sun NO2 measurements from Pandora instruments with a zenith-sky mode also applicable under cloudy conditions. In addition, attempts are made to also derive surface concentration estimates from total column observations. The general methodology is strongly inspired from the empirical zenith-sky mode developed for total ozone measurements by Dobson and Brewer spectrophotometers. Although the adopted approach implies many approximations not always well described or even identified (see detailed comments below), results are surprisingly good and certainly of interest for the ACP readership. I found the manuscript well written, concise and easy to read; also figures are of good quality and adequate in number and the appendices provide useful additional information. With

one exception, credit to existing literature is appropriate. I therefore recommend publication in ACP, after careful consideration for the comments and suggestions below.

**Specific comments**

Section 2.1.2, L. 20: what was the temperature used for the NO2 cross-section in the zenith-sky QDOAS retrievals? Is it consistent with the effective temperature assumed for the direct-sun retrieval  $(254.5^{\circ})$ ?

Section 2.1.2, L. 25: how was the NO2 residual amount in the reference spectrum determined here? Generally speaking, this paper lacks a proper analysis of the uncertainties. It would be useful to add a section describing the estimated uncertainties for the zenith-sky column and surface concentrations (which are new data products introduced in this study).

Section 3.1: The approach introduced for the zenith-sky AMF calculation is fully empirical and strongly inspired from the zenith-sky measurement mode used with Dobson and Brewer total ozone spectrophotometers. Basically the idea is to use simultaneous direct-sun and zenith-sky measurements to infer effective AMFs for the zenith-sky geometry. It is then assumed that the established relationship remains valid under moderately cloudy conditions (O4 is used to exclude thick diffusing clouds). I first note that the authors do not refer to the AMT publication by Tack et al. (2015) (https://www.atmosmeas-tech.net/8/2417/2015/) where a more physical approach to derive total and tropospheric NO2 columns from zenith-sky measurements is described. Second, I see a major drawback in the empirical approach used here, which is that the total AMF for zenith-sky measurements is expected to be a strong function of not only the solar zenith angle but also the tropospheric column itself. In first approximation, one can assume that the stratospheric AMF will mostly follow the solar geometry (geometrical AMF) while the PBL AMF is approximately constant and close to one at any solar position. In consequence, for intermediate and low sun conditions, the stratospheric and PBL AMFs differ quite strongly and the total AMF depends on the relative amount of NO2 present in the PBL and in the stratosphere. I think that the classification could be improved substantially by taking this dependence into account (probably within an iterative scheme). The dependence on the season accounts somehow for this effect (since it implicitly accounts for the seasonality of the stratospheric NO2 column), but only in a very crude way.

Section 3.1, L. 19: "... VCD\_Emp shows less SZA dependence than VCD\_DS..." -> I think that VCD\_NDACC is meant here instead of VCD\_DS

Section 3.1, last paragraph: note that the zenith-sky AMF could also be affected by aerosols present in PBL (together with NO2), due to their impact on the light path. Although the impact of aerosols is likely to be moderate, it certainly contribute to the uncertainty of the measurements and this should be mentioned.

Section 3.2, L. 5-10: the finding that zenith-sky columns despite their larger uncertainties (in comparison to direct-sun data) show a better agreement with satellite measurements is quite surprising and interesting. I am not convinced by the argument stating that the air mass sampled by zenith-sky measurements is more representative of the air mass sampled by the satellite than the direct-sun. Considering the size of typical OMI pixels (approx. 20x20 km2), one can argue that both direct-sun and zenith-sky measurements are more local in nature, and therefore maybe another explanation can be found. Could it be that direct-sun and zenith-sky measurements sample different meteorological conditions (e.g. different wind patterns), or maybe that zenith-sky are generally more homogeneously distributed around the overpass time of the satellite so that the average value becomes more representative? Please comment on these issues.

Section 4.1: the method used to convert NO2 column measurements into surface concentrations implies a lot of approximations/assumptions. The uncertainties associated to these assumptions should be better described. Equation 3 starts from the total NO2. For the zenith-sky case, this column already contains quite a large uncertainty (cf. pre-

СЗ

vious point). Then a correction for the stratospheric column and the free-tropospheric column is made. The stratospheric column is taken from photochemically-corrected stratospheric NO2 OMI measurements without any further verification. Can we exclude any possible systematic bias between OMI and ground-based measurements? Was there any attempt to verify that OMI and Pandora measurements do agree well under clean conditions? Also monthly mean OMI data are used. Can we safely neglect the day-to-day variability in the stratospheric NO2 content (e.g. in Spring it is known that transport patterns can produce short term variations of the stratospheric NO2)? As regards the free-tropospheric NO2 content, it is taken directly from model data. How large and uncertain is this contribution? Finally the last step is based on the assumption that the modelled column to surface concentration ratio is representative of the actual ratio. Is this assumption expected to be correct in all cases? What about the impact of the limited horizontal resolution of the model? In any case, I think that all these uncertainties are responsible for the large scatter of the correlations shown in Figure 6. As such they should be discussed with a little bit more of attention.

This being said, I agree that the average behaviors found (and illustrated in Figs. 9-11) are quite convincing and demonstrate well the potential of the data set.

Finally this study makes use of Pandora direct-sun completed by zenith-sky measurements. At no point in the paper, the potential of extending the data set with multi-axis measurements providing more information on the tropospheric NO2 is mentioned, although this would be a logical evolution for a follow up activity. Please consider adding this in the perspectives.

---

## Author Comment (AC1) · 4 Jul 2019

**Response to Referee #1:**

We thank referee #1 for their helpful comments. Our responses are given below in black with the referee's comments in blue. The new text in the modified manuscript is given in red (italicized).

**Referee #1:**

**General comments**

This paper describes the process to obtain the surface NO2 mixing ratio using the NO2 column from the ground-based Pandora observation. Some validations are performed by the comparison to the model outputs and in-situ measurement. The approaches look logical and the final products seem reasonable. Considering the rising importance of Pandora data for the air quality monitoring, publication of this work is useful to the research community and potential readers. But some more discussions should be included to improve this manuscript. Please refer to the comments below for the revision process.

**Specific comments**

P1, L27: Traffic is one of the important NO2 sources. Please include.

Done.

It is primarily emitted from combustion processes such as fossil fuel combustion (e.g., traffic, electricity generation from power plants) and biomass burning, as well as from lightning.  $NO_2$  is a nitrate aerosol precursor, and it also contributes to acid deposition and eutrophication (ECCC, 2016).

P2 L1-L12: Why is the total vertical column NO2 significantly treated first instead of insitu measurement. For air quality monitoring, in-situ measurement has been performed basically. Before the importance of Pandora observation is addressed, the necessity of ground-based remote sensing of trace gases should be stated first, in spite of the well-structured ground-based in-situ networks.

A new paragraph has been included in the Introduction to describe the current in situ measurements and the monitoring network (NAPS).

As surface NO2 concentrations are regulated by many environmental agencies (e.g., Environment and Climate Change Canada and U.S. Environmental Protection Agency), in situ NO2 measurements are commonly carried out by many national monitoring networks, such as the National Air Pollution Surveillance (NAPS, https://www.canada.ca/en/environment-climate-change/services/airpollution/monitoring-networks-data/national-air-pollution-program.html) network in Canada, which was established in 1969. The in situ methods used to measure surface NO2 have evolved over the years; for example, luminol chemiluminescence (e.g., Kelly et al., 1990; Maeda et al., 1980; Wendel et al., 1983), long-path differential optical absorption spectroscopy (e.g., Platt, 1994), photolytic conversion/chemiluminescence (e.g., Gao et al., 1994; Ryerson et al., 2000), and laser-induced fluorescence (e.g., Thornton et al., 2000) are all found to be reliable methods with an uncertainty within 10 % at the 1 ppbv and higher concentration levels (McClenny, 2000). Currently, the in situ approach used by NAPS for surface NO2 air quality monitoring is the photolytic conversion/chemiluminescence technique, which converts NO2 to NO and subsequently detects the NO by chemiluminescence reaction (McClenny, 2000; NRC, 1992). This in situ monitoring provides good measurements at ground level (0.4 ppbv accuracy), but NO2 is not uniformly mixed through the atmosphere, and not even within the atmospheric boundary layer due to emission and removal processes taking place at the surface.

P2 L8-9: The reason to use zenith-sky observations under the cloudy condition is not clear. Why zenith sky mode has more reliable than direct-sun mode under the cloudy condition?

The direct-sun mode requires unobscured sun to make reliable measurements. We have modified the text as follows:

Zenith-sky observations have been widely used for stratospheric ozone and NO2 observations, particularly under cloudy conditions when direct-sun measurements are unreliable (note that zenith-sky observations use scattered sunlight and are less sensitive to clouds, e.g., Zhao et al. (2019)).

P4 L20-21: Why do we need O4 retrieval? Still, some potential readers do not have any idea about the meaning of O4 retrieval.

Some background information about O4 is included in this new paragraph.

The oxygen collision complex  $(O_2)_2$  (referred here as  $O_4$ ), which is created by the collision of two oxygen molecules, has broadband absorptions from UV to near IR spectral ranges (Greenblatt et al., 1990; Platt and Stutz, 2008; Thalman and Volkamer, 2013).  $O_4$  is widely used as a reference gas by many DOAS applications to infer cloud and aerosol properties (e.g., Gielen et al., 2014; Wagner et al., 2004, 2014, 2016; Wang et al., 2015; Zhao et al., 2019).

P7 L1-2: In other Pandora paper (usually including Dr. Jay Herman in the author list), diurnal variation of stratospheric NO2 is not much considered. It will be more interesting (and even useful) to add some arguments compared with their approaches.

In short, there are two major concepts we must consider here, 1) the  $NO_2$  profile (weights between stratospheric and tropospheric  $NO_2$ ), and 2) the observation geometry (direct-sun or zenith-sky).

To calculate proper AMFs, there are two things that must be considered: the day-night difference due to photochemistry and the morning and evening peaks of NO2 due to local traffic. As discussed in the paper

(see Fig. A1b), the stratospheric diurnal variation contributes about 0.1 DU difference in total column NO2. Thus, this amount is large enough to matter for urban sites such as Toronto, and can be more significant for rural sites. In general, our conclusion is that an urban site with direct-sun observation should have smaller impact from the stratospheric diurnal variation. On the other hand, a rural site with zenith-sky observation should have a significant impact. This information has now been included in Appendix B as follows:

Please note that the strength of this bias is related to 1) the  $NO_2$  profile (weights between stratospheric and tropospheric  $NO_2$ ), and 2) the observation geometry (direct-sun or zenith-sky). In general, an urban site with direct-sun observation should have smaller impact from the stratospheric diurnal variation. On the other hand, a rural site with zenith-sky observation should have significant impact.

P8 L10-24: Notation is not clear. There are VCD\_DS and VCE\_ZS. Also, there are VCD\_EMP and VCD\_NDACC. VCD\_EMP (and VCD\_NDACC) is related to VCD\_DS or VCD\_ZS? Better notation or more description of these notations look required.

VCDEMP and VCDNDACC are both VCDs retrieved from zenith-sky observations. To make our notation clearer, we have changed VCDEMP and VCDNDACC to VCDZS-EMP and VCDZS-NADCC. The corresponding text has been changed as follows:

Total column NO2 can then be retrieved using Eqn. (1) and these two sets of AMFs, where the one based on empirical AMFs is referred to as  $VCD_{ZS-Emp}$  and the one based on NDACC AMFs is referred to as  $VCD_{ZS-Emp}$ NDACC. The RCD value used in the retrievals is 0.39 ± 0.01 DU, which is retrieved along with AMFZS-Emp (Appendix A).

In general, the VCDZS-Emp and VCDZS-NDACC performed as expected. Compared with VCDDS, the VCDZS-NDACC shows a -25% bias, while the VCDZS-Emp only shows a -4 % bias (indicated by the red lines on each panel and their slopes). In addition, VCDZS-Emp shows less SZA dependence than VCDZS-NDACC (see the increased bias for measurements made in larger SZA conditions in Figure 2b).

P10 L10-12: V\_ftrop is from the GEOS-chem simulations. How does GEOS-Chem consider the lightning NOx in the free-troposphere? In other words, GEOS-Chem can catch the free-tropospheric lightening NOx reasonably? It seems better to include some discussions about the estimation of lightning NOx in the free troposphere and even stratosphere, which is related to the final quality of C\_pan from the Pandora observation.

Lightning NOx emissions are computed as a function of cloud-top height, and are scaled globally as described by Sauvage et al. (2007) to match Optical Transient Detector/Lightning Imaging Sensor (OTD/LIS) climatological observations of lightning flashes. The global source is imposed to be 6 Tg(N) yr-1

(Martin et al., 2007). Higher NOx yields per flashes are used at mid-latitudes than in the tropics (Hudman et al., 2007). Different versions of the GEOS-CHEM model have been used extensively in the retrieval of tropospheric VCDs, and have been shown capable of simulating the vertical distributions of NO2 (e.g. Lamsal et al., 2008; Martin et al., 2002) and SO2 (e.g. Lee et al., 2009). Some of this information about the estimation of lightning NOx has now been included in Section 2.2.2.

The model has a detailed representation of tropospheric chemistry, including aerosols and their precursors (Park et al., 2004). In the simulation used in this study, a global lightning NOx source of 6 Tg N yr-1 (Martin et al., 2002) was imposed. Lightning NOx emissions are computed as a function of cloud-top height, and are scaled globally as described by Sauvage et al. (2007) to match Optical Transient Detector/Lightning Imaging Sensor (OTD/LIS) climatological observations of lightning flashes.

To provide some quality information about the final data product ( $C_{Pan}$ ), a detailed uncertainty estimation has been provided in a new Appendix D. In general, the contribution from the free tropospheric NO2 is limited compared to lower tropospheric and stratospheric NO2, as suggested by the following paragraph from Appendix D:

The estimated Pandora zenith-sky-based surface NO2 data have uncertainties from 4.8 to 6.5 ppbv. In Eqn. 8, the contributions of the  $V_{Pan}$ ,  $V_{Strat}$ ,  $V_{ftrop}$ , and  $R_{CV}$  terms to the total uncertainty are 36%, 2%, 0.3%, and 62 %, respectively. This result indicates that the uncertainty in the Pandora zenith-sky-based surface NO2 is dominated by the uncertainties of Pandora zenith-sky total column NO2 and modelled column-tosurface conversion ratio ( $R_{CV}$ ). However, please note that this uncertainty budget depends on the NO2 vertical distributions, and may vary from site to site; e.g., in Toronto, the tropospheric NO2 is typically 2-4 times higher than the stratospheric NO2, and thus, the contribution to uncertainty from  $V_{Pan}$  is much larger than the corresponding contributions from  $V_{Strat}$  and  $V_{ftrop}$ .

P10 L22-24: How about the year-to-year variation of conversion ratio? Is it larger or smaller than the diurnal variation of conversion ratio? If there is, it seems to include some discussion for how to treat or consider the year-to-year variation.

A year-to-year variation of conversion ratio is expected. However, the magnitude of this variation depends on the conditions of the site (e.g., local emission patterns, year round sunlight period). In other words, as the ratio in the look-up table is calculated using monthly mean (data period: April 2016 to December 2017), it is expected that for different period, this ratio will be different. For example, if the local emission patterns or meteorological conditions have significant changes for a site, the derived conversion ratio should reveal these changes. In general, the look-up table approach (Cpan-LUT) is aiming for a quick and near-real-time data delivery. Thus, for a given year, we recommend using a mean PSC-LUT that is calculated from model simulations of previous years. On the other hand, for the off-line data, the Cpan-model is the final, high-quality, year-specific data product that will be delivered to users. This information has now been included in this section:

In general, the look-up table approach ( $C_{pan-LUT}$ ) is aiming for a quick and near-real-time data delivery. Thus, to minimize year-to-year variation (e.g., from changing meteorological conditions or changing local emission patterns), for a given year we recommend using a mean PSC-LUT that is calculated from model simulations of previous years. On the other hand, the  $C_{pan-model}$  is the off-line, high-quality, year-specific data product that will be delivered for air quality research and other applications.

P11 L21-28: How to connect the results and discussions here to those in Fig. 4? The lesson from Fig. 4 seems that the ZS Pandora NO2 has better quality than the direct-sun Pandora NO2, right? But here both ZS and direct-sun Pandora NO2 shows consistent quality. Then what do we really learn from Fig. 4 in this work?

Please note the results in Fig. 4 are for total column NO2, while the discussion in page 11 is about Pandora derived surface NO2 concentrations. We also found slightly better agreement in Fig. 4 between Pandora zenith-sky NO2 VCD with OMI. However, due to the limited number of coincident data, we think the zenith-sky and direct-sun Pandora NO2 are of similar good quality. To further assess the quality, we need a longer period of observation and more coincident measurements. Some possible explanations are now included in Section 3.2.

The better correlation and lower bias for zenith-sky versus direct-sun might be a case of coincident errors, i.e., compare to Pandora direct-sun total column NO2, both OMI and Pandora zenith-sky total column NO2 underestimate the local NO2 at Toronto (see Figure 2). When taking into account the standard error of the fitting and the confidence level of R, the difference between zenith-sky and direct-sun data is not significant (i.e, in Fig. 4 from panels a to d, the slopes with standard error are 0.64  $\pm$  0.02, 0.67  $\pm$  0.02, 0.70  $\pm$  0.04, and 0.71  $\pm$  0.03; the 95% confidence intervals for R values are 0.45 to 0.63, 0.61 to 0.75, 0.43 to 0.77, and 0.60 to 0.86).

P12 L3-18 and Fig. 8: In this study, the validation is performed for the 2015-2017 period. Why only a month (APR 2017) is considered in this analysis? The better quality of ZS NO2 than direct-sun NO2 can be justified with this figure but whether the ZS NO2 always provide the reliable NO2 under the heavy clouds cannot be determined based on only 1-month situation. It seems necessary to deal with additional cloudy cases (at least in the supplements).

Fig. 8 is an example of the time series. It was chosen because of its good mixture of clear, moderate cloud, and heavy cloud conditions. The general performance of cloud filtering was included in Appendix C, which examines the quality of ZS  $NO_2$  in different cloud conditions (categorized by the enhancement of  $O_4$ ) for the period April 2016 to December 2017. We have included another month (October 2016) with good mixed conditions in Appendix C as another example.

Figure A4. Example of surface NO2 concentration time series in all conditions. The in situ, Pandora directsun (DS), and Pandora zenith-sky (ZS) surface NO2 concentrations are shown by different coloured dots. The total sky imager relative strength of direct-sun data are plotted as a colour-coded horizontal dot-line on the top area of each panel. For Pandora zenith-sky data, the measurements with enhanced  $O_4$  (heavy cloud indicator) are also labelled by green squares.

P13 L3-L20 and Figs. 9 and 10: In addition to the GEM-MACH model results, Pandora ZS and direct-sun NO2 products also show the higher standard deviation after sunrise compared to the time close to sunset. Considering the radiative extinction is more disturbed during the time close to sunrise and sunset, the higher standard deviation in the morning is not well understood. Why NO2 product in the morning has higher biases? This is also due to the propagation of GEM-MACH PBL uncertainty? (But authors mentioned that the surface NO2 from ZS Pandora observation is less dependent on the PBL height in P13 L29-30).

The standard deviations of Pandora ZS and DS in the morning are comparable to those of the in situ data. In Fig. 9, at 7:00 LST., in situ NO2 is  $14.9 \pm 9.3$  ppbv, while GEM-MACH, Pandora DS, and Pandora ZS NO2 are  $23.5 \pm 15.0$  ppbv,  $15.6 \pm 10.5$  ppbv, and  $15.2 \pm 6.8$  ppbv, respectively. To make it more clear, the shaded areas, which represent the 1 $\sigma$  envelope, have been modified to use dashed lines as boundaries.

---

## Author Comment (AC3) · 4 Jul 2019

Please find the revised manuscript in the Supplement.

Please also note the supplement to this comment:
https://www.atmos-chem-phys-discuss.net/acp-2018-1336/acp-2018-1336-AC3-supplement.pdf

---

## Author Response (AR1)

**Response to Referee #1:**

We thank referee #1 for their helpful comments. Our responses are given below in black with the referee's comments in blue. The new text in the modified manuscript is given in red (italicized).

Referee #1:

General comments
This paper describes the process to obtain the surface NO2 mixing ratio using the NO2 column from the ground-based Pandora observation. Some validations are performed by the comparison to the model outputs and in-situ measurement. The approaches look logical and the final products seem reasonable. Considering the rising importance of Pandora data for the air quality monitoring, publication of this work is useful to the research community and potential readers. But some more discussions should be included to improve this manuscript. Please refer to the comments below for the revision process.

Specific comments

P1, L27: Traffic is one of the important NO2 sources. Please include.

Done.

*It is primarily emitted from combustion processes such as fossil fuel combustion (e.g., traffic, electricity generation from power plants) and biomass burning, as well as from lightning. $NO_2$ is a nitrate aerosol precursor, and it also contributes to acid deposition and eutrophication (ECCC, 2016).*

P2 L1-L12: Why is the total vertical column NO2 significantly treated first instead of insitu measurement. For air quality monitoring, in-situ measurement has been performed basically. Before the importance of Pandora observation is addressed, the necessity of ground-based remote sensing of trace gases should be stated first, in spite of the well-structured ground-based in-situ networks.

A new paragraph has been included in the Introduction to describe the current in situ measurements and the monitoring network (NAPS).

*As surface $NO_2$ concentrations are regulated by many environmental agencies (e.g., Environment and Climate Change Canada and U.S. Environmental Protection Agency), in situ $NO_2$ measurements are commonly carried out by many national monitoring networks, such as the National Air Pollution Surveillance (NAPS, https://www.canada.ca/en/environment-climate-change/services/air-pollution/monitoring-networks-data/national-air-pollution-program.html) network in Canada, which was established in 1969. The in situ methods used to measure surface $NO_2$ have evolved over the years; for example, luminol chemiluminescence (e.g., Kelly et al., 1990; Maeda et al., 1980; Wendel et al., 1983), long-path differential optical absorption spectroscopy (e.g., Platt, 1994), photolytic*

*conversion/chemiluminescence (e.g., Gao et al., 1994; Ryerson et al., 2000), and laser-induced fluorescence (e.g., Thornton et al., 2000) are all found to be reliable methods with an uncertainty within 10 % at the 1 ppbv and higher concentration levels (McClenny, 2000). Currently, the in situ approach used by NAPS for surface $NO_2$ air quality monitoring is the photolytic conversion/chemiluminescence technique, which converts $NO_2$ to NO and subsequently detects the NO by chemiluminescence reaction (McClenny, 2000; NRC, 1992). This in situ monitoring provides good measurements at ground level (0.4 ppbv accuracy), but $NO_2$ is not uniformly mixed through the atmosphere, and not even within the atmospheric boundary layer due to emission and removal processes taking place at the surface.*

P2 L8-9: The reason to use zenith-sky observations under the cloudy condition is not clear. Why zenith sky mode has more reliable than direct-sun mode under the cloudy condition?

The direct-sun mode requires unobscured sun to make reliable measurements. We have modified the text as follows:

*Zenith-sky observations have been widely used for stratospheric ozone and $NO_2$ observations, particularly under cloudy conditions when direct-sun measurements are unreliable (note that zenith-sky observations use scattered sunlight and are less sensitive to clouds, e.g., Zhao et al. (2019)).*

P4 L20-21: Why do we need O4 retrieval? Still, some potential readers do not have any idea about the meaning of O4 retrieval.

Some background information about $O_4$ is included in this new paragraph.

*The oxygen collision complex $(O_2)_2$ (referred here as $O_4$), which is created by the collision of two oxygen molecules, has broadband absorptions from UV to near IR spectral ranges (Greenblatt et al., 1990; Platt and Stutz, 2008; Thalman and Volkamer, 2013). $O_4$ is widely used as a reference gas by many DOAS applications to infer cloud and aerosol properties (e.g., Gielen et al., 2014; Wagner et al., 2004, 2014, 2016; Wang et al., 2015; Zhao et al., 2019).*

P7 L1-2: In other Pandora paper (usually including Dr. Jay Herman in the author list), diurnal variation of stratospheric NO2 is not much considered. It will be more interesting (and even useful) to add some arguments compared with their approaches.

In short, there are two major concepts we must consider here, 1) the $NO_2$ profile (weights between stratospheric and tropospheric $NO_2$), and 2) the observation geometry (direct-sun or zenith-sky).

To calculate proper AMFs, there are two things that must be considered: the day-night difference due to photochemistry and the morning and evening peaks of $NO_2$ due to local traffic. As discussed in the paper

(see Fig. A1b), the stratospheric diurnal variation contributes about 0.1 DU difference in total column $NO_2$. Thus, this amount is large enough to matter for urban sites such as Toronto, and can be more significant for rural sites. In general, our conclusion is that an urban site with direct-sun observation should have smaller impact from the stratospheric diurnal variation. On the other hand, a rural site with zenith-sky observation should have a significant impact. This information has now been included in Appendix B as follows:

*Please note that the strength of this bias is related to 1) the $NO_2$ profile (weights between stratospheric and tropospheric $NO_2$), and 2) the observation geometry (direct-sun or zenith-sky). In general, an urban site with direct-sun observation should have smaller impact from the stratospheric diurnal variation. On the other hand, a rural site with zenith-sky observation should have significant impact.*

P8 L10-24: Notation is not clear. There are VCD_DS and VCE_ZS. Also, there are VCD_EMP and VCD_NDACC. VCD_EMP (and VCD_NDACC) is related to VCD_DS or VCD_ZS? Better notation or more description of these notations look required.

$VCD_{EMP}$ and $VCD_{NDACC}$ are both VCDs retrieved from zenith-sky observations. To make our notation clearer, we have changed $VCD_{EMP}$ and $VCD_{NDACC}$ to $VCD_{ZS-EMP}$ and $VCD_{ZS-NADCC}$. The corresponding text has been changed as follows:

*Total column $NO_2$ can then be retrieved using Eqn. (1) and these two sets of AMFs, where the one based on empirical AMFs is referred to as $VCD_{ZS-Emp}$ and the one based on NDACC AMFs is referred to as $VCD_{ZS-NDACC}$. The RCD value used in the retrievals is 0.39 ± 0.01 DU, which is retrieved along with $AMF_{ZS-Emp}$ (Appendix A).*

*In general, the $VCD_{ZS-Emp}$ and $VCD_{ZS-NDACC}$ performed as expected. Compared with $VCD_{DS}$, the $VCD_{ZS-NDACC}$ shows a -25% bias, while the $VCD_{ZS-Emp}$ only shows a -4 % bias (indicated by the red lines on each panel and their slopes). In addition, $VCD_{ZS-Emp}$ shows less SZA dependence than $VCD_{ZS-NDACC}$ (see the increased bias for measurements made in larger SZA conditions in Figure 2b).*

P10 L10-12: V_ftrop is from the GEOS-chem simulations. How does GEOS-Chem consider the lightning NOx in the free-troposphere? In other words, GEOS-Chem can catch the free-tropospheric lightening NOx reasonably? It seems better to include some discussions about the estimation of lightning NOx in the free troposphere and even stratosphere, which is related to the final quality of C_pan from the Pandora observation.

Lightning $NO_x$ emissions are computed as a function of cloud-top height, and are scaled globally as described by Sauvage et al. (2007) to match Optical Transient Detector/Lightning Imaging Sensor (OTD/LIS) climatological observations of lightning flashes. The global source is imposed to be 6 Tg(N) yr$^{-1}$

(Martin et al., 2007). Higher $NO_x$ yields per flashes are used at mid-latitudes than in the tropics (Hudman et al., 2007). Different versions of the GEOS-CHEM model have been used extensively in the retrieval of tropospheric VCDs, and have been shown capable of simulating the vertical distributions of $NO_2$ (e.g. Lamsal et al., 2008; Martin et al., 2002) and $SO_2$ (e.g. Lee et al., 2009). Some of this information about the estimation of lightning NOx has now been included in Section 2.2.2.

*The model has a detailed representation of tropospheric chemistry, including aerosols and their precursors (Park et al., 2004). In the simulation used in this study, a global lightning $NO_x$ source of 6 Tg N $yr^{-1}$ (Martin et al., 2002) was imposed.* *Lightning $NO_x$ emissions are computed as a function of cloud-top height, and are scaled globally as described by Sauvage et al. (2007) to match Optical Transient Detector/Lightning Imaging Sensor (OTD/LIS) climatological observations of lightning flashes.*

To provide some quality information about the final data product ($C_{Pan}$), a detailed uncertainty estimation has been provided in a new Appendix D. In general, the contribution from the free tropospheric $NO_2$ is limited compared to lower tropospheric and stratospheric $NO_2$, as suggested by the following paragraph from Appendix D:

*The estimated Pandora zenith-sky-based surface $NO_2$ data have uncertainties from 4.8 to 6.5 ppbv. In Eqn. 8, the contributions of the $V_{Pan}$, $V_{Strat}$, $V_{ftrop}$, and $R_{CV}$ terms to the total uncertainty are 36%, 2%, 0.3%, and 62 %, respectively. This result indicates that the uncertainty in the Pandora zenith-sky-based surface $NO_2$ is dominated by the uncertainties of Pandora zenith-sky total column $NO_2$ and modelled column-to-surface conversion ratio ($R_{CV}$). However, please note that this uncertainty budget depends on the $NO_2$ vertical distributions, and may vary from site to site; e.g., in Toronto, the tropospheric $NO_2$ is typically 2-4 times higher than the stratospheric $NO_2$, and thus, the contribution to uncertainty from $V_{Pan}$ is much larger than the corresponding contributions from $V_{Strat}$ and $V_{ftrop}$.*

P10 L22-24: How about the year-to-year variation of conversion ratio? Is it larger or smaller than the diurnal variation of conversion ratio? If there is, it seems to include some discussion for how to treat or consider the year-to-year variation.

A year-to-year variation of conversion ratio is expected. However, the magnitude of this variation depends on the conditions of the site (e.g., local emission patterns, year round sunlight period). In other words, as the ratio in the look-up table is calculated using monthly mean (data period: April 2016 to December 2017), it is expected that for different period, this ratio will be different. For example, if the local emission patterns or meteorological conditions have significant changes for a site, the derived conversion ratio should reveal these changes. In general, the look-up table approach ($C_{pan-LUT}$) is aiming for a quick and near-real-time data delivery. Thus, for a given year, we recommend using a mean PSC-LUT that is calculated from model simulations of previous years. On the other hand, for the off-line data, the $C_{pan-model}$ is the final, high-quality, year-specific data product that will be delivered to users. This information has now been included in this section:

*In general, the look-up table approach ($C_{pan-LUT}$) is aiming for a quick and near-real-time data delivery. Thus, to minimize year-to-year variation (e.g., from changing meteorological conditions or changing local emission patterns), for a given year we recommend using a mean PSC-LUT that is calculated from model simulations of previous years. On the other hand, the $C_{pan-model}$ is the off-line, high-quality, year-specific data product that will be delivered for air quality research and other applications.*

P11 L21-28: How to connect the results and discussions here to those in Fig. 4? The lesson from Fig. 4 seems that the ZS Pandora NO2 has better quality than the direct-sun Pandora NO2, right? But here both ZS and direct-sun Pandora NO2 shows consistent quality. Then what do we really learn from Fig. 4 in this work?

Please note the results in Fig. 4 are for total column $NO_2$, while the discussion in page 11 is about Pandora derived surface $NO_2$ concentrations. We also found slightly better agreement in Fig. 4 between Pandora zenith-sky $NO_2$ VCD with OMI. However, due to the limited number of coincident data, we think the zenith-sky and direct-sun Pandora $NO_2$ are of similar good quality. To further assess the quality, we need a longer period of observation and more coincident measurements. Some possible explanations are now included in Section 3.2.

*The better correlation and lower bias for zenith-sky versus direct-sun might be a case of coincident errors, i.e., compare to Pandora direct-sun total column $NO_2$, both OMI and Pandora zenith-sky total column $NO_2$ underestimate the local $NO_2$ at Toronto (see Figure 2). When taking into account the standard error of the fitting and the confidence level of R, the difference between zenith-sky and direct-sun data is not significant (i.e, in Fig. 4 from panels a to d, the slopes with standard error are 0.64 ± 0.02, 0.67 ± 0.02, 0.70 ± 0.04, and 0.71 ± 0.03; the 95% confidence intervals for R values are 0.45 to 0.63, 0.61 to 0.75, 0.43 to 0.77, and 0.60 to 0.86).*

P12 L3-18 and Fig. 8: In this study, the validation is performed for the 2015-2017 period. Why only a month (APR 2017) is considered in this analysis? The better quality of ZS NO2 than direct-sun NO2 can be justified with this figure but whether the ZS NO2 always provide the reliable NO2 under the heavy clouds cannot be determined based on only 1-month situation. It seems necessary to deal with additional cloudy cases (at least in the supplements).

Fig. 8 is an example of the time series. It was chosen because of its good mixture of clear, moderate cloud, and heavy cloud conditions. The general performance of cloud filtering was included in Appendix C, which examines the quality of ZS $NO_2$ in different cloud conditions (categorized by the enhancement of $O_4$) for the period April 2016 to December 2017. We have included another month (October 2016) with good mixed conditions in Appendix C as another example.

[Figure]

*Figure A4. Example of surface NO$_2$ concentration time series in all conditions. The in situ, Pandora direct-sun (DS), and Pandora zenith-sky (ZS) surface NO$_2$ concentrations are shown by different coloured dots. The total sky imager relative strength of direct-sun data are plotted as a colour-coded horizontal dot-line on the top area of each panel. For Pandora zenith-sky data, the measurements with enhanced O$_4$ (heavy cloud indicator) are also labelled by green squares.*

P13 L3-L20 and Figs. 9 and 10: In addition to the GEM-MACH model results, Pandora ZS and direct-sun NO2 products also show the higher standard deviation after sunrise compared to the time close to sunset. Considering the radiative extinction is more disturbed during the time close to sunrise and sunset, the higher standard deviation in the morning is not well understood. Why NO2 product in the morning has higher biases? This is also due to the propagation of GEM-MACH PBL uncertainty? (But authors mentioned that the surface NO2 from ZS Pandora observation is less dependent on the PBL height in P13 L29-30).

The standard deviations of Pandora ZS and DS in the morning are comparable to those of the in situ data. In Fig. 9, at 7:00 LST., in situ NO$_2$ is 14.9 ± 9.3 ppbv, while GEM-MACH, Pandora DS, and Pandora ZS NO$_2$ are 23.5 ± 15.0 ppbv, 15.6 ± 10.5 ppbv, and 15.2 ± 6.8 ppbv, respectively. To make it more clear, the shaded areas, which represent the 1σ envelope, have been modified to use dashed lines as boundaries.

[Figure]

For evening conditions, in Fig. 9, at 17:00 LST, in situ $NO_2$ is 7.3 ± 5.8 ppbv, while GEM-MACH, Pandora DS, and Pandora ZS $NO_2$ are 5.6 ± 5.0 ppbv, 3.6 ± 2.6 ppbv, and 5.2 ± 3.4 ppbv, respectively. The larger morning standard deviations in all observations and modelled data are related to different $NO_2$ emission patterns in the mornings (i.e., work-days and weekend). The lower PBL in the mornings also enhanced the impacts from emissions compared to afternoon conditions. This information has now been included in the paragraph:

*For example, at 7:00 LST, in situ $NO_2$ is 14.9 ± 9.3 ppbv, while GEM-MACH, Pandora DS, and Pandra ZS $NO_2$ are 23.5 ± 15.0 ppbv, 15.6 ± 10.5 ppbv, and 15.2 ± 6.8 ppbv, respectively. At 17:00 LST, in situ $NO_2$ is 7.3 ± 5.8 ppbv, while GEM-MACH, Pandora DS, and Pandora ZS $NO_2$ are 5.6 ± 5.0 ppbv, 3.6 ± 2.6 ppbv, and 5.2 ± 3.4 ppbv, respectively. The larger standard deviations in the morning are due to the datasets not being divided into work-days and weekends.*

Technical corrections

P6 L2: two times -> twice

Done.

P7 L30-32: The meaning of this statement is not clear to me. Please more clarify the explanation.

It has been modified to be clearer.

*Instead of finding the link between zenith-sky* *spectral* *intensity and total column* *values (i.e., following the Brewer and Dobson zenith-sky total ozone retrieval method),* *deriving empirical zenith-sky AMFs for Pandora zenith-sky measurements is more straightforward since Pandora zenith-sky spectra can be analyzed to produce $NO_2$ dSCDs.*

P8 L10-12: This statement looks redundant, just a repetition of statements above.

The statement in P8 L10-12 is not redundant. The notation of this part has been modified to make this clearer:

*If we make an assumption that the coincident direct-sun (DS) and zenith-sky (ZS) measurements sampled the same air mass, then the empirical zenith-sky AMFs* *(referred to here as $AMF_{ZS-Emp}$)* *can be calculated by assuming $VCD_{DS} = VCD_{ZS}$, which gives*

$$VCD_{DS}(SZA) = \frac{dSCD_{ZS}(SZA) + RCD_{ZS}}{AMF_{ZS-Emp}(SZA)} . \quad (2)$$

*Next, we can use nearly-coincident $VCD_{DS}$ and $dSCD_{ZS}$ in a multi-non-linear regression to retrieve $AMF_{ZS-Emp}$ and $RCD_{ZS}$ together. To ensure the quality of the retrieved $AMF_{ZS-Emp}$, only high quality direct-sun total column $NO_2$ data are used with SZA < 75°. Details about the empirical zenith-sky AMF calculation are shown in Appendix A.*

*Figure 1 shows a comparison of the empirical zenith-sky AMFs and NDACC AMFs (calculated for the Toronto measurements). Total column $NO_2$ can then be retrieved using Eqn. (1) and these two sets of AMFs, where the one based on empirical AMFs is referred to as $VCD_{ZS-Emp}$ and the one based on NDACC AMFs is referred to as $VCD_{ZS-NDACC}$. The RCD value used in the retrievals is 0.39 ± 0.01 DU, which is retrieved along with $AMF_{ZS-Emp}$ (Appendix A).*

P11 L10: This result (slightly lower correlation) -> This slightly lower correlation Figs. 4, 7, and A2: If a slope value is the only significant value, then 'slope = \*\*\*' looks better expression. If authors would show the information with the format of 'y = slope x', it seems better to add intercept values together, to have the original regression equation (the equation looks not completed without intercept values)

The fitting that we used here is not a simple linear fitting, but a non-linear fit with intercept set to zero (this information was provided in the captions of Figs. 4, 7, and A2). The reason to force the intercept to be zero is to provide a meaningful value to represent the bias between two datasets. The assumption we made here is that if atmospheric $NO_2$ is zero, then any instrument should measure zero $NO_2$. In other words, the assumption is that there should be no additive systematic bias between instruments (i.e., drifting from zero), but only multiplicative systematic bias exist. In reality, although this assumption might not be true, for any instrument that we know, if it has a drift from zero (an additive systematic

bias) we first correct this drifting before using its data. In other words, although the intercept value in simple linear fitting could be an indicator for multiplicative systematic bias between instruments, the value itself depends on which instrument we trust as truth (i.e., which dataset is used as y, and which dataset is used as x). Thus, the intercept is not really a meaningful indicator to show here.

In short, the fittings (intercept forced to be zero) are used to reveal the bias between two datasets, and the equations shown on those figures are complete.

Figure captions: It seems that all figure captions can be more clarified to better describe the figures. Please check and correct once more before the submission of revised manuscript (Now they are somewhat confused. Fig. 6 and 7 are examples. What really figure b and c imply? Direct description of figure a, b, c looks better to read).

The captions have been modified as requested. The implications of Panels (b) and (c) in each figure (Figures 6 and 7) are included at the end of Section 4.2.

*Figure 6. Modelled and Pandora zenith-sky surface $NO_2$ vs. in situ $NO_2$. (a) shows the GEM-MACH modelled surface $NO_2$ data vs. in situ $NO_2$; (b) and (c) show the Pandora zenith-sky (ZS) surface $NO_2$ data vs. in situ $NO_2$. The Pandora ZS surface $NO_2$ data in (b) and (c) are derived using the hourly modelled conversion ratio and the monthly PSC-LUT, respectively. (d) to (f) are histograms corresponding to the data in (a) to (c). On each scatter plot, the red line is the linear fit with intercept set to 0 and the black line is the one-to-one line. The scatter plots are colour-coded by the normalized density of the points.*

*Figure 7. Modelled and Pandora direct-sun surface $NO_2$ vs. in situ $NO_2$. (a) shows the GEM-MACH modelled surface $NO_2$ data vs. in situ $NO_2$; (b) and (c) show the Pandora direct-sun (DS) surface $NO_2$ data vs. in situ $NO_2$. The Pandora DS surface $NO_2$ data in (b) and (c) are derived using the hourly modelled conversion ratio and the monthly PSC-LUT, respectively. (d) to (f) are histograms corresponding to the data in (a) to (c). On each scatter plot, the red line is the linear fit with intercept set to 0 and the black line is the one-to-one line. The scatter plots are colour-coded by the normalized density of the points.*

*The good consistency between $C_{pan-model}$, and $C_{pan-LUT}$ implies that two versions of Pandora surface $NO_2$ data can be delivered in the future, i.e., an off-line version that relies on hourly inputs from the model, and a near-real-time version that only needs a pre-calculated LUT.*

Time period: Please clearly suggest the time period of each analysis (at least in each figure caption). I think each analysis has a different time period of analysis, which is a little complex to me. While I can accept this difference, the clear time information should be provided to help the readers' understanding.

Done. Captions of Figures 2, 4, and 6 to 10 are modified to include the period of each analysis.

**Reference**

[revised manuscript text omitted]

**Response to Referee #2:**

We thank referee #2 for their helpful comments. Our responses are given below in black with the referee's comments in blue. The new text in the modified manuscript is given in red (italicized).

**Referee #2:**

General comments
This study presents a new approach to extend direct-sun NO2 measurements from Pandora instruments with a zenith-sky mode also applicable under cloudy conditions. In addition, attempts are made to also derive surface concentration estimates from total column observations. The general methodology is strongly inspired from the empirical zenith-sky mode developed for total ozone measurements by Dobson and Brewer spectrophotometers. Although the adopted approach implies many approximations not always well described or even identified (see detailed comments below), results are surprisingly good and certainly of interest for the ACP readership. I found the manuscript well written, concise and easy to read; also figures are of good quality and adequate in number and the appendices provide useful additional information. With one exception, credit to existing literature is appropriate. I therefore recommend publication in ACP, after careful consideration for the comments and suggestions below.

Specific comments

Section 2.1.2, L. 20: what was the temperature used for the NO2 cross-section in the zenith-sky QDOAS retrievals? Is it consistent with the effective temperature assumed for the direct-sun retrieval (254.5 K)?

The effective temperatures for $NO_2$ and ozone have now been included in the manuscript. The temperature used for the ZS $NO_2$ was 254.5 K, which is consistent with the effective temperature used for the direct-sun retrieval.

*Cross sections of NO$_2$ at an effective temperature of 254.5 K (Vandaele et al., 1998), ozone at an effective temperature of 223 K (Bogumil et al., 2003), H$_2$O (Rothman et al., 2005), O$_4$ (Hermans et al., 2003), and Ring (Chance and Spurr, 1997) are all fitted; a fifth-order polynomial and a first-order linear offset are also included in the DOAS analysis.*

Section 2.1.2, L. 25: how was the NO2 residual amount in the reference spectrum determined here? Generally speaking, this paper lacks a proper analysis of the uncertainties. It would be useful to add a section describing the estimated uncertainties for the zenith-sky column and surface concentrations (which are new data products introduced in this study).

The residual amount in the reference spectrum (RCD) was determined by using the proposed multi-non-linear regressions (see Appendix A). In short, the RCD was retrieved along with the empirical AMFs.

From the multi-non-linear model, the standard fitting error of RCD is 0.01 DU. This information was included in Section 3.1.

*The RCD value used in the retrievals is 0.39 ± 0.01 DU, which is retrieved along with $AMF_{ZS\text{-}Emp}$ (Appendix A).*

In addition, as suggested by the referee the following new section (Appendix D) has been included that describes the estimated uncertainties for the zenith-sky column and surface concentrations:

**D. Uncertainty estimation**

[revised manuscript text omitted]

Section 3.1: The approach introduced for the zenith-sky AMF calculation is fully empirical and strongly inspired from the zenith-sky measurement mode used with Dobson and Brewer total ozone spectrophotometers. Basically the idea is to use simultaneous direct-sun and zenith-sky measurements to infer effective AMFs for the zenith-sky geometry. It is then assumed that the established relationship remains valid under moderately cloudy conditions (O4 is used to exclude thick diffusing clouds). I first note that the authors do not refer to the AMT publication by Tack et al. (2015) (https://www.atmosmeas- tech.net/8/2417/2015/) where a more physical approach to derive total and

tropospheric NO2 columns from zenith-sky measurements is described. Second, I see a major drawback in the empirical approach used here, which is that the total AMF for zenith-sky measurements is expected to be a strong function of not only the solar zenith angle but also the tropospheric column itself. In first approximation, one can assume that the stratospheric AMF will mostly follow the solar geometry (geometrical AMF) while the PBL AMF is approximately constant and close to one at any solar position. In consequence, for intermediate and low sun conditions, the stratospheric and PBL AMFs differ quite strongly and the total AMF depends on the relative amount of NO2 present in the PBL and in the stratosphere. I think that the classification could be improved substantially by taking this dependence into account (probably within an iterative scheme). The dependence on the season accounts somehow for this effect (since it implicitly accounts for the seasonality of the stratospheric NO2 column), but only in a very crude way.

The information from Tack et al. (2015) has now been included in Section 3.1 and this publication has been included in the References section:

*In Tack et al. (2015), a more sophisticated four-step approach to derive total and tropospheric $NO_2$ columns from zenith-sky measurements was proposed, which involves using a RTM to calculate appropriate tropospheric AMFs. However, due to benefits from using the high-quality Pandora direct-sun total column $NO_2$ measurements, this work took a different but simple and robust approach to derive zenith-sky total column $NO_2$.*

We thank the referee for their insightful suggestions about improving the empirical AMF calculations. The current empirical AMFs are limited to high and intermediate sun conditions (i.e., SZA< 75°). In the future, when deriving low-sun empirical ZS AMFs, we will try the suggested iterative scheme to account for the stronger influence of PBL $NO_2$. We think this suggestion is valuable, and we have now included this information in the manuscript (Appendix A) as follows:

*In addition, the current empirical AMFs are limited to high and intermediate sun conditions (i.e., SZA< 75°). For low-sun conditions, the total AMF for zenith-sky measurements is expected to be a strong function of not only the SZA, but also the tropospheric column itself. Thus, for future work to derive low-sun empirical zenith-sky AMFs, the stronger influence of PBL $NO_2$ has to be accounted for (i.e., the geometry form AMFs are not enough).*

Section 3.1, L. 19: " : : : VCD_Emp shows less SZA dependence than VCD_DS: : : " -> I think that VCD_NDACC is meant here instead of VCD_DS

Corrected.

*In addition, $VCD_{Emp}$ shows less SZA dependence than $VCD_{NDACC}$ (see the increased bias for measurements made in larger SZA conditions in Figure 2b).*

Section 3.1, last paragraph: note that the zenith-sky AMF could also be affected by aerosols present in PBL (together with NO2), due to their impact on the light path. Although the impact of aerosols is likely to be moderate, it certainly contribute to the uncertainty of the measurements and this should be mentioned.

Thank you for this comment. The following new text has been included in Section 3.1, last paragraph:

*The derived zenith-sky total column $NO_2$ values are affected by both clouds and aerosols due to their impact on the light path. The presence of clouds and aerosols contributes to the uncertainty of the measurements. However, the impact of aerosols is expected to be moderate in most cases compared to that of clouds (e.g., Hendrick et al., 2011; Tack et al., 2015). Thus, this work has focused on evaluating the impact from clouds.*

Section 3.2, L. 5-10: the finding that zenith-sky columns despite their larger uncertainties (in comparison to direct-sun data) show a better agreement with satellite measurements is quite surprising and interesting. I am not convinced by the argument stating that the air mass sampled by zenith-sky measurements is more representative of the air mass sampled by the satellite than the direct-sun. Considering the size of typical OMI pixels (approx. 20x20 km2), one can argue that both direct-sun and zenith-sky measurements are more local in nature, and therefore maybe another explanation can be found. Could it be that direct-sun and zenith-sky measurements sample different meteorological conditions (e.g. different wind patterns), or maybe that zenith-sky are generally more homogeneously distributed around the overpass time of the satellite so that the average value becomes more representative? Please comment on these issues.

We were not able to find a solid reason for this finding (zenith-sky measurements have better agreement with OMI). The measurement frequency was given in Section 2.1.1 and the coincidence criteria were provided in Section 3.2. We agree with the referee that both direct-sun and zenith-sky measurements are more local when compared with OMI measurements. However, with the coincidence criteria used (± 30 min), we are not sure that meteorological conditions are the key factor. Also, we did not average the ground-based data. The nearest (in time) measurement that was within ± 30 min of OMI overpass time was used (see Section 3.2). Therefore, it is not an averaging issue. In addition, when taking the standard error of the fitting and the confidence level of R into account, the difference between zenith-sky and direct-sun data is not significant (i.e., in Fig. 4 from panels a to d, the slopes with standard error are 0.64 ± 0.02, 0.67 ± 0.02, 0.70 ± 0.04, and 0.71 ± 0.03; the 95% confidence interval for R values are 0.45 to 0.63, 0.61 to 0.75, 0.43 to 0.77, and 0.60 to 0.86). In general, we think it is probably a case of coincident error, i.e., compared to Pandora DS, both OMI and Pandora ZS underestimate the local $NO_2$ at Toronto (see Section 3.2 and Figure 2). More discussion of this issue has now been included in Section 3.2:

*The better correlation and lower bias for zenith-sky versus direct-sun might be a case of coincident error, i.e., compared to Pandora direct-sun, both OMI and Pandora zenith-sky total column $NO_2$ underestimate*

*the local NO$_2$ at Toronto (see Figure 2). When taking into account the standard error of the fitting and the confidence level of R, the difference between zenith-sky and direct-sun data is not significant (i.e, in Fig. 4 from panesl a to d, the slopes with standard error are 0.64 ± 0.02, 0.67 ± 0.02, 0.70 ± 0.04, and 0.71 ± 0.03; the 95% confidence intervals for R values are 0.45 to 0.63, 0.61 to 0.75, 0.43 to 0.77, and 0.60 to 0.86).*

Section 4.1: the method used to convert NO2 column measurements into surface concentrations implies a lot of approximations/assumptions. The uncertainties associated to these assumptions should be better described. Equation 3 starts from the total NO2. For the zenith-sky case, this column already contains quite a large uncertainty (cf. previous point). Then a correction for the stratospheric column and the free tropospheric column is made. The stratospheric column is taken from photochemically-corrected stratospheric NO2 OMI measurements without any further verification. Can we exclude any possible systematic bias between OMI and ground-based measurements? Was there any attempt to verify that OMI and Pandora measurements do agree well under clean conditions? Also monthly mean OMI data are used. Can we safely neglect the day-to-day variability in the stratospheric NO2 content (e.g. in Spring it is known that transport patterns can produce short term variations of the stratospheric NO2)?

There could be a systematic bias between OMI and ground-based data but most likely the bias found at Toronto is related to tropospheric NO$_2$. Thus, we decided to assess the OMI stratospheric NO$_2$ data and used it in the algorithm. In this work, we tested the algorithm using both OMI daily and monthly mean stratospheric NO$_2$, and we did not find any significant differences in the derived surface NO$_2$. The reason is that for Toronto, tropospheric NO$_2$ accounts for 73 % of the total column amounts on average (see Section 3.1) around local noon. During rush hours, the percentage of tropospheric NO$_2$ should be even higher. Also, when using OMI daily stratospheric NO$_2$, we need to interpolate the data for cloud days, which will also increase uncertainty. One of the reasons that we decided to use monthly mean OMI stratospheric NO$_2$ was also to reveal the robustness of the method (i.e., not highly dependent on satellite measurements). Some of this information has now been included in Appendix B:

*Note that the strength of this bias is related to 1) the NO$_2$ profile (weights between stratospheric and tropospheric NO$_2$), and 2) the observation geometry (direct-sun or zenith-sky). In general, an urban site with direct-sun observation should have less impact from the stratospheric diurnal variation. On the other hand, a rural site with zenith-sky observation should have significant impact.*

As regards the free-tropospheric NO$_2$ content, it is taken directly from model data. How large and uncertain is this contribution? Finally the last step is based on the assumption that the modelled column to surface concentration ratio is representative of the actual ratio. Is this assumption expected to be correct in all cases? What about the impact of the limited horizontal resolution of the model? In any case, I think that all these uncertainties are responsible for the large scatter of the correlations shown in Figure 6. As such they should be discussed with a little bit more of attention. This being said, I agree that

the average behaviors found (and illustrated in Figs. 9-11) are quite convincing and demonstrate well the potential of the data set.

The current method is sophisticated in that outputs from several different chemical transport models (with different strengths) were included. Currently, we are working to provide estimate of uncertainties of the final data products (Pandora surface $NO_2$). Based on the Toronto datasets, the mean of free tropospheric $NO_2$ is 0.03 ± 0.01 DU (mean ± 1σ) (GEOS-Chem), while the boundary layer $NO_2$ is 0.37 ± 0.29 DU (GEM-MACH), and stratospheric $NO_2$ is 0.10 ± 0.02 DU (OMI monthly mean stratospheric $NO_2$, with Pratmo to account for diurnal variation). Thus, the uncertainty contributions from GEOS-Chem, OMI, and Pratmo are much less than the uncertainty contribution from GEM-MACH. An uncertainty estimation section is now included as Appendix D (see the reply to previous comments), which addresses the uncertainties associated with the model inputs.

The assumption that the modelled column to surface concentration ratio is representative of the actual ratio has challenges, especially for shallow boundary layer conditions (e.g., Figure 11). However, we think the model can better represent the real conditions for most cases (taking emissions, atmospheric dynamics, and chemistry into account), when comparing to other existing methods (which simply assume that $NO_2$ is uniformly mixed through the entire PBL). The current GEM-MACH operational forecast model has spatial resolution of 10 km × 10 km (information was provided in Section 2.2.1). Based on our current results, the model shows some bias compared to in situ data, which could have contributions from the spatial resolution.

Finally this study makes use of Pandora direct-sun completed by zenith-sky measurements. At no point in the paper, the potential of extending the data set with multi-axis measurements providing more information on the tropospheric NO2 is mentioned, although this would be a logical evolution for a follow up activity. Please consider adding this in the perspectives.

Thank you for this comment. Text about the potential of Pandora multi-axis measurements has been added to the Conclusions section.

[revised manuscript text omitted]